# Peroxiredoxin 5 regulates osteogenic differentiation through interaction with hnRNPK during bone regeneration

**Eunjin Cho[1†], Xiangguo Che[2†], Mary Jasmin Ang[3], Seongmin Cheon[4,5], Jinkyung Lee[1], Kwang Soo Kim[6], Chang Hoon Lee[7], Sang-Yeop Lee[8], Hee-Young Yang[9], Changjong Moon[10], Chungoo Park[4], Je-Yong Choi[2], Tae-Hoon Lee[1]\***

[1]Department of Oral Biochemistry, Korea Mouse Phenotype Center (KMPC), Dental Science Research Institute, School of Dentistry, Chonnam National University, Gwangju, Republic of Korea; [2]Department of Biochemistry and Cell Biology, BK21 Plus KNU Biomedical Convergence Program, Skeletal Diseases Analysis Center, Korea Mouse Phenotyping Center (KMPC), School of Medicine, Kyungpook National University, Daegu, Republic of Korea; [3]Department of Basic Veterinary Sciences, College of Veterinary Medicine, University of the Philippines Los Baños, Los Baños, Philippines; [4]School of Biological Sciences and Technology, Chonnam National University, Gwangju, Republic of Korea; [5]Proteomics Core Facility, Biomedical Research Institute, Seoul National University Hospital, Seoul, Republic of Korea; [6]Department of Microbiology, Department of Molecular Medicine (BK21plus), Chonnam National University Medical School, Gwangju, Republic of Korea; [7]Therapeutic & Biotechnology Division, Drug Discovery Platform Research Center, Research Institute of Chemical Technology (KRICT), Daejeon, Republic of Korea; [8]Research Center for Bioconvergence Analysis, Korea Basic Science Institute, Ochang, Republic of Korea; [9]Preclinical Research Center, Daegu-Gyeongbuk Medical Innovation Foundation, Daegu, Republic of Korea; [10]Department of Veterinary Anatomy and Animal Behavior, College of Veterinary Medicine and BK21 FOUR Program, Chonnam National University, Gwangju, Republic of Korea

**\*For correspondence:**
thlee83@chonnam.ac.kr

[†]These authors contributed equally to this work

**Competing interest:** The authors declare that no competing interests exist.

**Abstract** Peroxiredoxin 5 (Prdx5) is involved in pathophysiological regulation via the stress-induced cellular response. However, its function in the bone remains largely unknown. Here, we show that Prdx5 is involved in osteoclast and osteoblast differentiation, resulting in osteoporotic phenotypes in *Prdx5* knockout (*Prdx5*^Ko) male mice. To investigate the function of Prdx5 in the bone, osteoblasts were analyzed through immunoprecipation (IP) and liquid chromatography combined with tandem mass spectrometry (LC–MS/MS) methods, while osteoclasts were analyzed through RNA-sequencing. Heterogeneous nuclear ribonucleoprotein K (hnRNPK) was identified as a potential binding partner of Prdx5 during osteoblast differentiation in vitro. Prdx5 acts as a negative regulator of hnRNPK-mediated osteocalcin (*Bglap*) expression. In addition, transcriptomic analysis revealed that in vitro differentiated osteoclasts from the bone marrow-derived macrophages of *Prdx5*^Ko mice showed enhanced expression of several osteoclast-related genes. These findings indicate that Prdx5 might contribute to the maintenance of bone homeostasis by regulating osteoblast differentiation. This study proposes a new function of Prdx5 in bone remodeling that may be used in developing therapeutic strategies for bone diseases.

## Editor's evaluation

Peroxiredoxin 5 regulates osteogenic differentiation via interaction with hnRNPK during bone regeneration is an important study as the fundamental role of Peroxiredoxin 5 has been established in bone regeneration. The study is compelling with experimentally establishing the role of Peroxiredoxin- 5 in osteoblast, and osteoclast along with in vivo studies using Prdx5 knockout mice to establish the functional role of Prdx in bone homeostasis and further as a therapeutic target.

## Introduction

Bone is remodeled through continuous replacement of old tissues by new tissues; 5–7% of bone mass is recycled every week (*Sims and Walsh, 2012*). Bone remodeling is a highly complex process between bone deposition or production by osteoblasts and bone resorption by osteoclasts, which are responsible for the breakdown of old bone tissues (*Knothe Tate et al., 2004*; *Yang et al., 2020*). Osteoblasts originating from mesenchymal stromal cells undergo the Wingless-Int and bone morphogenic protein (BMP) pathways to terminally differentiate into osteocytes (*Zaidi, 2007*). First, mature osteoblasts secrete type I collagen and bone extracellular matrix proteins, and then osteocytes are surrounded by collagen matrix (*Kim et al., 2020a*). Osteoclasts are derived from hematopoietic stem cells upon stimulation of the receptor activator of nuclear factor kappa-B ligand (RANKL) and macrophage colony-stimulating factor (M-CSF). Mature osteoclasts resorb bone via secretion of acid and dissolving enzymes (*Lopes et al., 2018*). The balance between osteoclast and osteoblast activities is critical for bone remodeling; an imbalance between these cells can lead to various bone disorders (*Weitzmann and Ofotokun, 2016*).

World Health Organization defines osteoporosis as a reduction 2.5 standard deviations (SDs) or more below the mean peak bone mineral density (BMD) in young adults, resulting in an increase in bone fragility and fractures (*Compston et al., 2019*). Prevalence of osteoporosis increases with age because of the changes in hormones, vitamin D, growth factors, and bone cell distribution (*Zanker and Duque, 2019*). Globally, 34% of post-menopausal women and 17% of men were suffer from osteoporosis (*Laird et al., 2023*; *Pavone et al., 2017*). Bone remodeling increases in older women owing to deficiency in estrogen, which thins the trabecular lining and reduced the cortical thickness (*Compston et al., 2019*). In men, osteoporotic fractures increase with age because of a greater bending strength and a lower moment of inertia resulting from trabecular thinning, and increased bone fragility with greater endocortical expansion and continuous periosteal apposition (*Lambert et al., 2011*). In general, both enhanced osteoclastic resorption and reduced osteoblastic bone formation lead to bone loss, resulting in osteoporosis (*Iqbal and Zaidi, 2021*).

Peroxiredoxins (Prdxs) are a large superfamily of antioxidant enzymes that reduce peroxides (*Rhee, 2016*). They are classified as 2-Cys (Prdx1–5) and 1-Cys (Prdx6) based on their conserved cysteine residues (*Seong et al., 2021*). They protect cells from oxidative stress (*Lee et al., 2020*; *Rhee, 2016*). Prdx6 inhibits bone formation in newborn mice (*Park et al., 2019*). Thioredoxin-1 induces osteoclast differentiation, which is suppressed by glutathione peroxidase-1 and Prdx1 (*Lean et al., 2004*). Prdx5 acts as a mitochondrial antioxidant and regulates ciliogenesis, adipogenesis, and fibrogenesis (*Choi et al., 2019*; *Ji et al., 2019*; *Kim et al., 2018*). Furthermore, it ameliorates obesity-induced non-alcoholic fatty liver disease by modulating the mitochondrial reactive oxygen species (ROS) (*Kim et al., 2020b*). From a biochemical perspective, Prdx5 regulates the activation of cyclin-dependent kinase 5 and $Ca^{2+}$/calcineurin-Drp1, Jak2–Stat5 modulation during pathogenic conditions via antioxidant activity, and protein–protein interactions (*Choi et al., 2013*; *Chung et al., 2010*; *Park et al., 2016*; *Park et al., 2017*; *Yang et al., 2010*). However, the role of Prdx5 in bone remodeling has not yet been studied.

Here, we examine the role of Prdxs in osteoblast and osteoclast differentiation in vitro. Interestingly, Prdx5 expression significantly altered during bone cell differentiation. Therefore, we also define the role of Prdx5 in the bone using *Prdx5*-deficient (*Prdx5*Ko) mice. To determine the interacting partners of Prdx5 during osteogenesis, we performed liquid chromatography with tandem mass spectrometry (LC–MS/MS) analysis. RNA-sequencing (RNA-seq) analysis showed a significant increase in the expression of osteoclast-related genes in the osteoclasts differentiated from bone marrow-derived macrophages (BMMs) of *Prdx5*Ko mice compared to that in the osteoclasts of wild-type (WT) mice. Our findings indicate a new role of Prdx5 in bone biology, that is, Prdx5 homeostasis is critical for bone

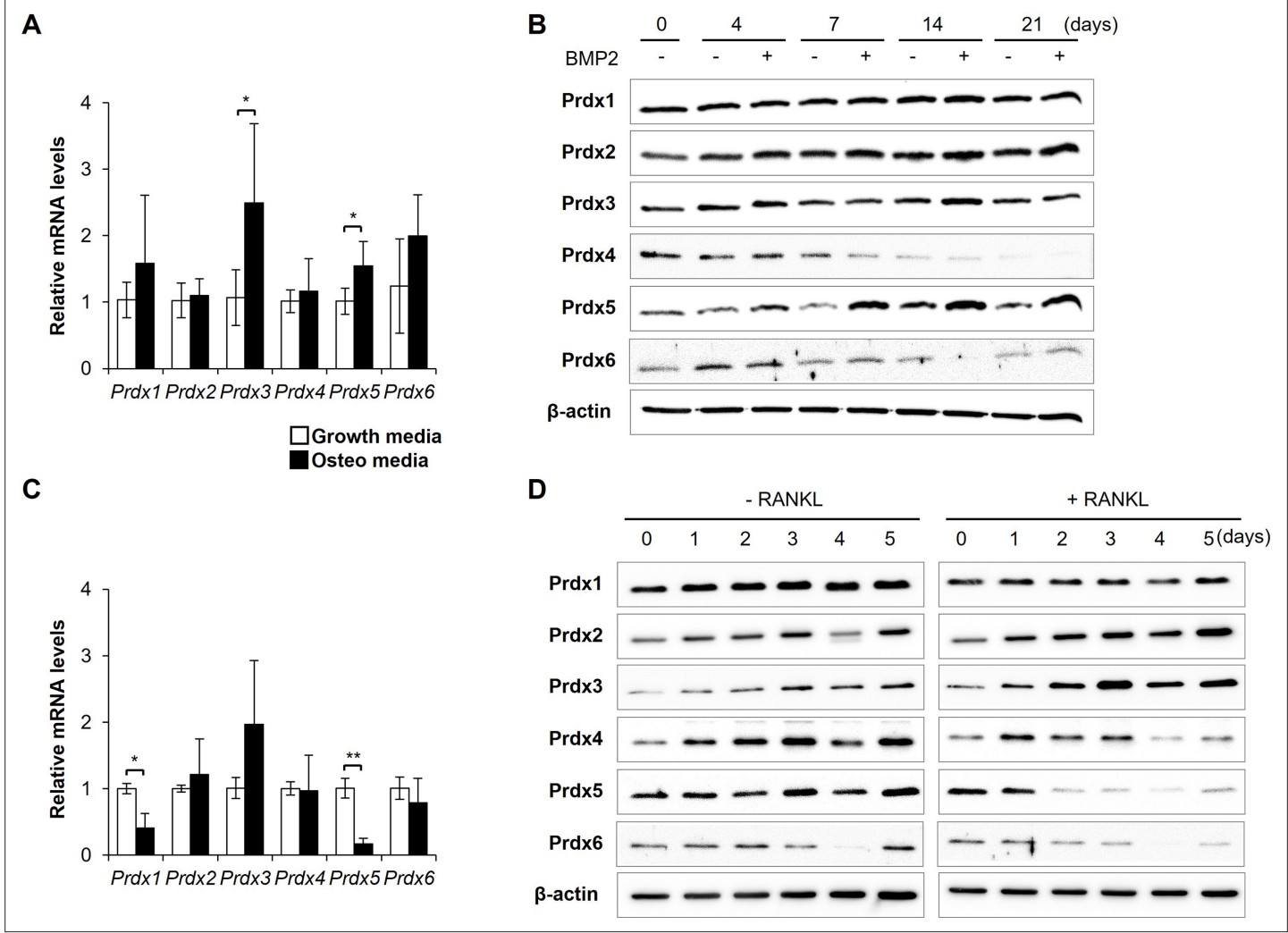

**Figure 1.** Peroxiredoxin 5 (Prdx5) expression is controlled during bone cell differentiation. (**A**) mRNA expression of Prdxs was determined in osteoblasts, using quantitative reverse transcription-PCR (qRT-PCR), on day 7 after bone morphogenic protein 2 (BMP2) stimulation. (**B**) Protein levels of Prdxs in osteoblasts were determined via western blotting. (**C**) mRNA levels of Prdxs were determined in osteoclasts on day 3 after receptor activator of nuclear factor kappa-B ligand (RANKL) stimulation. (**D**) Protein levels of Prdxs in osteoclasts were determined using western blotting. Growth media comprised only serum. Osteo-media comprised either BMP2 (**A, B**) or RANKL (**C, D**) for osteoclasts and osteoblasts, respectively. Graph depicts mean ± SD. *p<0.05, **p<0.01 via an unpaired two-tailed Student's t-test between growth media vs. osteo-media samples (n=3).

The online version of this article includes the following source data for figure 1:

**Source data 1.** Full blots for *Figure 1B and D*.

**Source data 2.** Original blots for *Figure 1B and D*.

remodeling. Therefore, Prdx5 may prove useful for understanding and preventing various osteoporotic disorders involving osteoclast activity.

## Results

### Prdx5 is controlled during bone cell differentiation

To elucidate whether Prdxs function in bone remodeling, we characterized the expression of all Prdxs (Prdx1–6) during osteoclast and osteoblast differentiation in vitro (*Figure 1*). First, calvaria-derived pre-osteoblasts were differentiated into osteoblasts by stimulation with bone morphogenic protein 2 (BMP2). The mRNA levels of *Prdx3* and *Prdx5* significantly elevated upon BMP2 stimulation (*Figure 1A*). During osteoblast differentiation, BMP2 stimulation increased the Prdx2 and Prdx5

protein expressions (*Figure 1B*). To explore the function of Prdxs in osteoclast differentiation, BMMs were differentiated into osteoclasts by treatment with RANKL. The mRNA levels of *Prdx1* and *Prdx5* significantly reduced (*Figure 1C*). Prdx4 and Prdx5 protein expressions were altered by RANKL treatment (*Figure 1D*). Interestingly, Prdx5 levels decreased during osteoclastogenesis but increased during osteogenesis, indicating a correlation between mRNA and protein expression. Therefore, we investigated Prdx5 as a potential regulator of bone remodeling.

## Abnormal expression of Prdx5 modulates osteoblastogenesis and osteoclastogenesis in vitro

To clarify the roles of Prdx5 in osteoblast and osteoclast differentiation, we thoroughly examined its expression in vitro. BMP2 stimulation increased the *Prdx5* mRNA expression on days 4 and 7 and decreased it on day 14 (*Figure 2A*). However, the Prdx5 protein expression was continuously upregulated till day 14. We isolated the precursor cells from *Prdx5*$^{Ko}$ mice and examined osteoblast differentiation using alkaline phosphatase (ALP, *Alpl*) staining. Osteoblast differentiation was strongly inhibited in *Prdx5*$^{Ko}$ cells on day 7 (*Figure 2B*). To examine the expression of osteoblast-specific genes, we performed quantitative reverse transcription-PCR (qRT-PCR) on day 7 after BMP2 administration (*Figure 2C*). The mRNA levels of Runt-related transcription factor 2 (*Runx2*), *Alpl*, and *Bglap* increased. However, the upregulation in *Prdx5*$^{Ko}$ cells was significantly reduced compared to that in WT.

Next, we analyzed the Prdx5 expression during osteoclast differentiation after administering the RANKL and M-CSF (*Figure 2D–F*). The Prdx5 expression decreased from the first day of osteoclastogenesis (*Figure 2D*). The efficacy of osteoclast differentiation was examined in the BMMs from *Prdx5*$^{Ko}$ mice (*Figure 2E*). To determine the number of differentiated osteoclasts, tartrate-resistant acid phosphatase (TRAP, *Acp5*) staining was performed. Interestingly, the *Prdx5*$^{Ko}$ BMMs showed a two-fold increase in TRAP-positive areas compared to that observed with the WT. Osteoclasts become multinucleated giant cells through cell–cell fusion and acquire bone resorption activity (***Kodama and Kaito, 2020***). Therefore, we measured the number of nuclei in a TRAP-positive cell as an indicator of cell fusion. *Prdx5*$^{Ko}$ cells had fewer nuclei than in WT. During osteoclastogenesis, the levels of *Acp5* and cathepsin K (*Ctsk*) remarkably increase in mature osteoclasts. The transcription factor c-Fos regulates the nuclear factor of activated T-cells cytoplasmic 1 (NFATc1)-mediated signaling pathways (***Nagy and Penninger, 2015***; ***Yang and Karsenty, 2002***). The mRNA levels of *Acp5* and *Ctsk* reduced in *Prdx5*$^{Ko}$ cells on day 3 (*Figure 2F*) but increased up to the levels typically found in WT on days 4 and 5 (*Figure 2—figure supplement 1*). These data suggest that, in *Prdx5*$^{Ko}$, BMMs develop osteoclasts at a slower rate than that in WT. These differences do not change even at the maturation stage of osteoclasts in vitro.

To determine whether Prdx5 deficiency increases ROS levels, we determined ROS levels in cultured osteoblasts and osteoclasts obtained from *Prdx5*$^{Ko}$ and WT mice (*Figure 2—figure supplement 2*). The pre-osteoblasts from *Prdx5*$^{Ko}$ mice showed slightly reduced ROS levels than those from WT. In osteoblasts, ROS levels increased in WT cells after BMP2 stimulation, whereas they remained at the precursor levels in *Prdx5*$^{Ko}$ cells. However, ROS levels did not change upon RANKL stimulation during osteoclast differentiation in both WT and *Prdx5*$^{Ko}$ cells. We elucidated the mRNA levels of nicotinamide adenine dinucleotide phosphate oxidase (Nox) and their cytoplasmic subunits (*Figure 2—figure supplement 2*). The *Cybb* (*Nox2*), *Ncf4* (*p40*), *Ncf1* (*p47*), and *Rac2* mRNA levels in the osteoblast precursor cells obtained from *Prdx5*$^{Ko}$ mice were lower than their levels in cells from WT. These results suggest that the ROS levels decreased in the *Prdx5*$^{Ko}$ osteoblast cells. The *Cybb*, *Noxo1*, *Cyba* (*p22*), *Ncf4*, and *Ncf2* (*p67*) mRNA levels in the osteoclasts from *Prdx5*$^{Ko}$ were lower than that in the cells from WT. Overall, *Prdx5*$^{Ko}$-derived osteoblasts showed reduced ROS and mRNA levels of Nox compounds, while osteoclasts from *Prdx5*$^{Ko}$ cells revealed slightly suppressed mRNA levels. These results suggest that ROS are not significantly involved in Prdx5-mediated osteoblast or osteoclast differentiation.

Additionally, we transfected Prdx5 and their cysteine mutants, replacing each cysteine residues with serine into *Prdx5*$^{Ko}$ pre-osteoblasts, and then tested osteoblast differentiation (*Figure 2—figure supplement 3*). Prdx5 is an atypical 2-Cys Prdx, which forms an intramolecular disulfide bond between Cys$^{48}$ and Cys$^{152}$ (***Seo et al., 2000***). After transfection with expression vectors encoding Prdx5, Prdx5-C48S, or/and -C152S, ALP assays were performed to examine osteoblast differentiation. Prdx5-deficient cells showed an increase in ALP-positive cells by Prdx5 transfection. However, their recovery effects remained the same even in Prdx5 cysteine mutant-transfected cells.

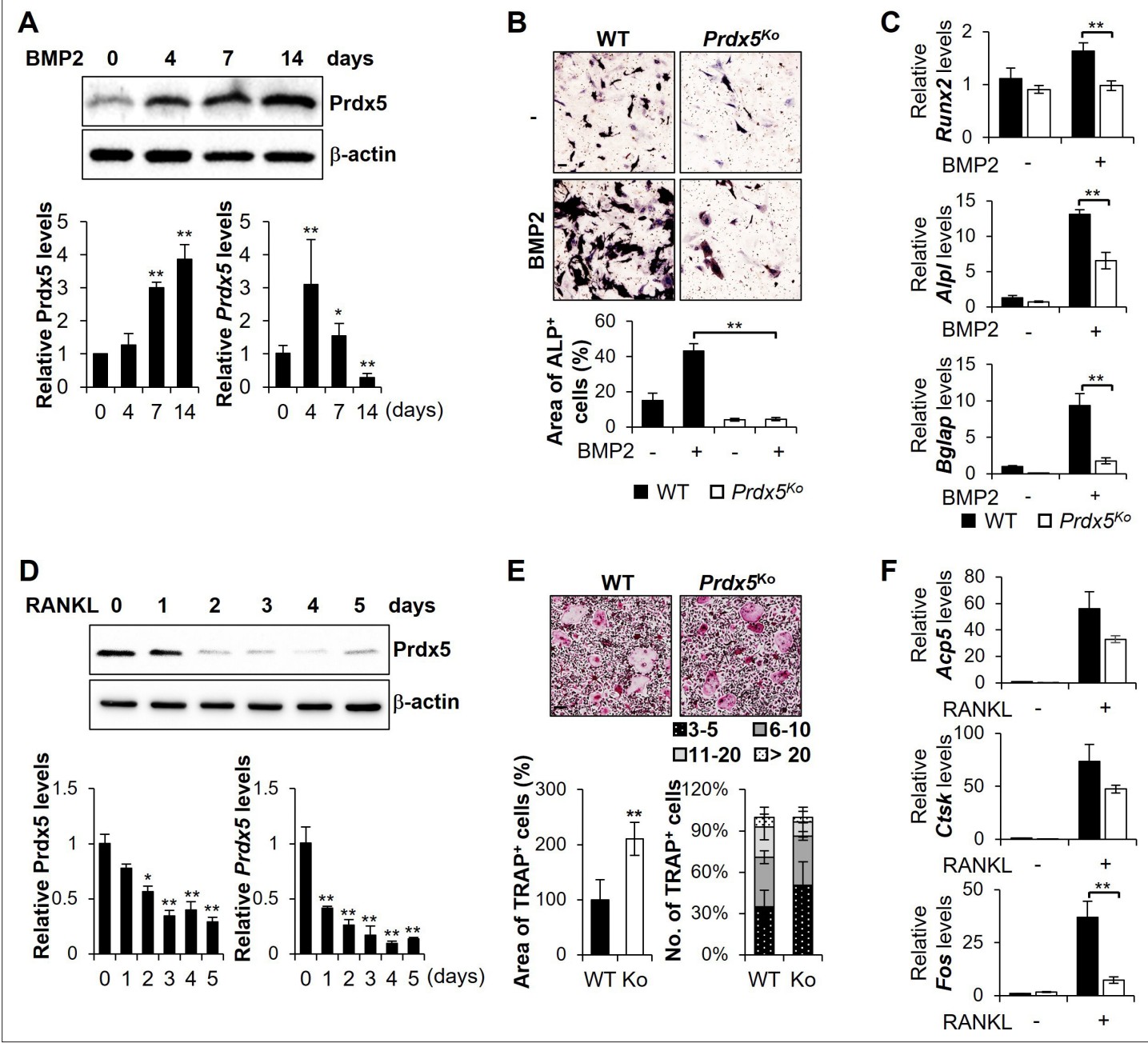

**Figure 2.** Abnormal expression of peroxiredoxin 5 (Prdx5) modulates osteoblastogenesis and osteoclastogenesis in vitro. (**A, B, C**) Mouse calvaria-derived pre-osteoblasts were differentiated into osteoblasts through bone morphogenic protein 2 (BMP2) stimulation for indicated time periods. (**A**) Western blotting (upper and left bottom) and quantitative reverse transcription-PCR (qRT-PCR) (right bottom) were performed to determine Prdx5 expression during osteoblastogenesis. (**B**) Pre-osteoblasts were isolated from wild-type (WT) and *Prdx5*[Ko] mice and then differentiated into osteoblasts for 7 days. Alkaline phosphatase (ALP) staining was performed to determine the osteoblasts, and the area of ALP-positive cells was measured using the ImageJ software. (**C**) qRT-PCR was performed to determine osteogenic gene expression on day 7. (**D, E, F**) Bone marrow-derived macrophages (BMMs) were differentiated into osteoclasts through macrophage colony-stimulating factor (M-CSF) and receptor activator of nuclear factor kappa-B ligand (RANKL) stimulation for indicated time periods. (**D**) Western blotting (upper and left bottom) and qRT-PCR (right bottom) were performed to determine Prdx5 expression during osteoclastogenesis. (**E**) BMMs were isolated from WT and *Prdx5*[Ko] mice and then differentiated into osteoclasts for 4 days. Tartrate-resistant acid phosphatase (TRAP) staining was performed to determine the osteoclasts. The area of TRAP-positive cells was measured, and the number of multinucleated cells harboring the indicated nuclei was counted. (**F**) qRT-PCR was performed to determine the expression of osteoclast-related genes. Graph depicts mean ± SEM. *p<0.05, **p <0.01 via an unpaired two-tailed Student's t-test compared to control (0) or WT.

The online version of this article includes the following source data and figure supplement(s) for figure 2:

**Source data 1.** Full blots for *Figure 2A and D*.

*Figure 2 continued on next page*

*Figure 2 continued*

**Source data 2.** Original blots for *Figure 2A and D*.

**Figure supplement 1.** Quantitative reverse transcription-PCR (qRT-PCR) was performed to determine the expression of osteoclast-related genes during osteoclastogenesis.

**Figure supplement 2.** Reactive oxygen species (ROS) levels are not altered in peroxiredoxin 5 (*Prdx5*)-deficient osteoblasts.

**Figure supplement 3.** Osteoblast differentiation is not altered by peroxiredoxin 5 (Prdx5) cysteine mutants.

## *Prdx5*<sup>Ko</sup> male mice show enhanced osteoporotic phenotypes

To determine the role of Prdx5 in bone remodeling in vivo, we analyzed bone parameters in *Prdx5*<sup>Ko</sup> mice (*Figure 3*). Micro-CT analysis of distal femurs showed that *Prdx5*<sup>Ko</sup> male mice had low BMD and trabecular number (Tb. N) and an increased trabecular bone space (Tb. Sp) compared to those in the WT littermates (*Figure 3B*). Additionally, *Prdx5*<sup>Ko</sup> male mice showed reduced trabecular volume (Tb. V) and thickness (Tb. Th), which suggested reduced trabecular bone formation in *Prdx5*<sup>Ko</sup> male mice compared to that in WT. To determine bone-related cytokine levels in the serum, RANKL, osteoprotegerin (OPG), and BMP2 levels were examined (*Figure 3C*). In the *Prdx5*<sup>Ko</sup> male mice, RANKL levels increased by 1.5-fold compared to those in WT. However, BMP2 levels were not altered in *Prdx5*<sup>Ko</sup> male mice. Therefore, osteoporosis-like phenotypes in *Prdx5*<sup>Ko</sup> male mice were mediated by an increase in RANKL expression.

We confirmed the osteogenic potential in mouse femurs stained with TRAP and ALP, which are the markers of osteoclasts and osteoblasts, respectively (*Figure 3D*). The number of total TRAP-positive cells was not altered in *Prdx5*<sup>Ko</sup> male mice. Because *Prdx5*<sup>Ko</sup> male mice have less trabecular bone volume (*Figure 3B*), we measured the ratio of osteoclast and bone surfaces. *Prdx5*<sup>Ko</sup> male mice showed higher osteoclast surface ratios than that of WT (*Figure 3E*). The total number of ALP-positive cells reduced in *Prdx5*<sup>Ko</sup> male mice; however, the reduction was not statistically significant. Altogether, *Prdx5*<sup>Ko</sup> male mice showed increased number of osteoclasts in the femurs. These osteoporotic phenotypes were not observed in female mice (*Figure 3—figure supplement 1*). *Prdx5*<sup>Ko</sup> females showed no differences in the BMD, bone volume, and trabecular bone thickness and space. Therefore, we examined bone parameters in an ovariectomy-induced osteoporosis mouse model (OVX) (*Figure 3—figure supplement 1*). Micro-CT analysis revealed that OVX mice displayed significantly lower Tb. V and Tb. N than that in sham mice; however, no significant differences were observed between WT and *Prdx5*<sup>Ko</sup> female mice.

To evaluate whether male-specific osteoporotic phenotypes were related with testosterone, testosterone levels were examined from both the WT and *Prdx5*<sup>Ko</sup> male mice sera (*Figure 3—figure supplement 2*). Interestingly, WT and *Prdx5*<sup>Ko</sup> male mice reveled similar testosterone levels before puberty (4–8 weeks); their levels did not enhance at 12 weeks in the *Prdx5*<sup>Ko</sup> males as much as it did in the WT mice. Additionally, androgen receptor (AR) expression levels were determined in WT and *Prdx5*<sup>Ko</sup> male mice. The mRNA and protein levels were downregulated in osteoblasts from *Prdx5*<sup>Ko</sup> mice, although their expression levels were significantly decreased by BMP2 stimulation in WT. In osteoclasts, mRNA and protein levels of the AR were not altered between WT and *Prdx5*<sup>Ko</sup> mice. However, RANKL stimulation decreased the AR expression levels. These data suggested that osteoporotic phenotypes in *Prdx5*<sup>Ko</sup> mice are related with reduced testosterone and AR levels in osteoblasts.

## Limited bone remodeling activities in *Prdx5*<sup>Ko</sup> male mice

To determine the bone remodeling activity, we examined the bone turnover rates in *Prdx5*<sup>Ko</sup> male mice. First, we confirmed osteoblast function using trichrome staining and by conducting dynamic bone histomorphometry analysis in vivo (*Figure 4—figure supplement 1*). Trichrome staining revealed lower osteoid volume per bone volume in *Prdx5*<sup>Ko</sup> male mice than in WT, indicating reduced bone modeling in *Prdx5*-deficient mice. In *Prdx5*<sup>Ko</sup> mice, a lower width between calcein and Alizarin Red S labeling and lower mineral apposition rate (MAR) in the trabecular bone were observed than those in WT mice. However, cortical bone revealed no such alteration. Thus, *Prdx5*<sup>Ko</sup> male mice exhibited reduced bone turnover parameters, which indicated the suppression of newly formed bone tissue in the trabecular bone.

Next, to test osteogenic potential in vivo, we analyzed the osteogenic healing capacity using the calvarial defect model in *Prdx5*<sup>Ko</sup> male mice and their WT littermates (*Figure 4*). After the calvarial

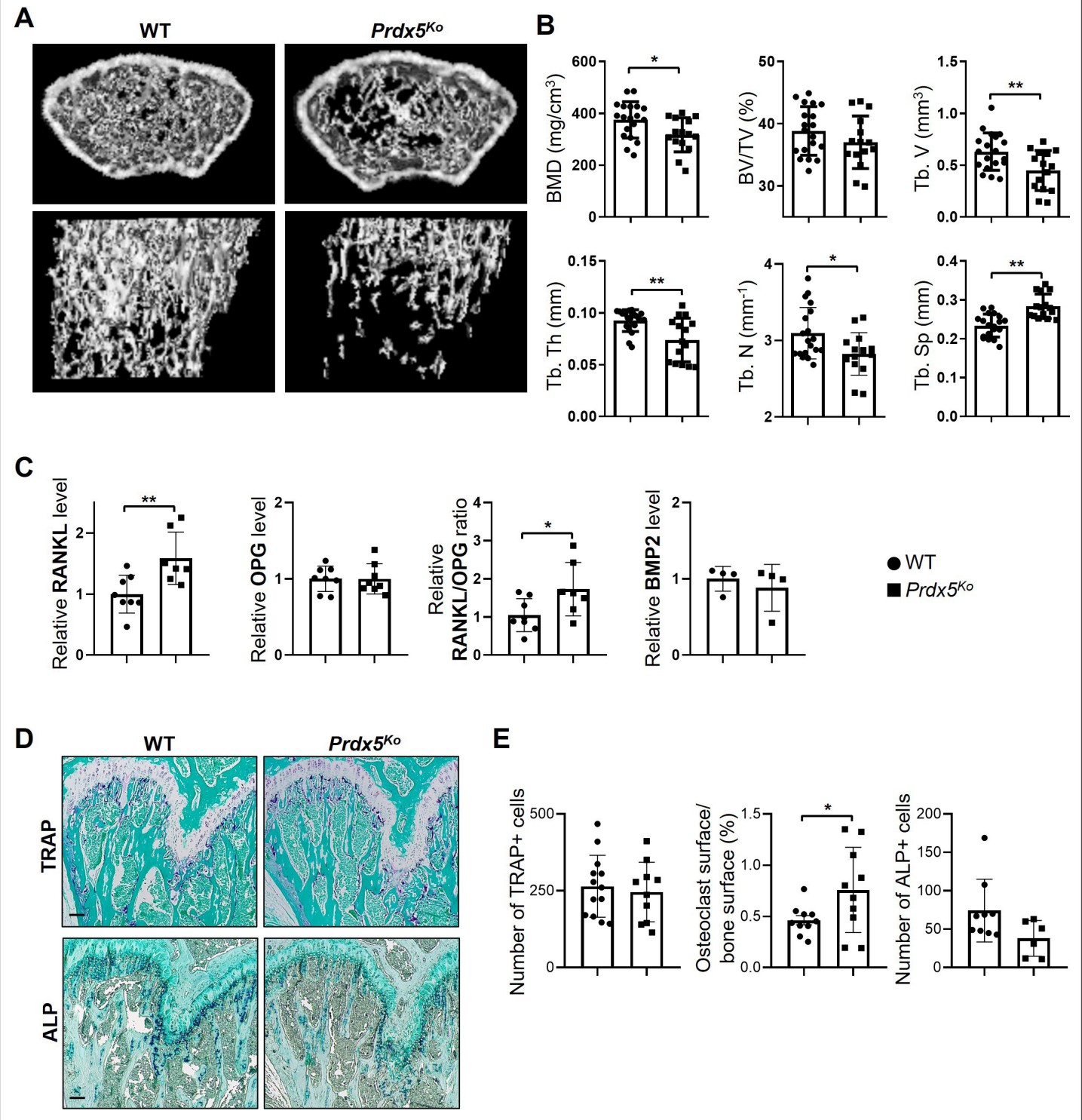

**Figure 3.** *Prdx5*^Ko^ male mice show enhanced osteoporotic phenotypes. (**A**) Micro-CT images of femurs from 12-week-old wild-type (WT) and *Prdx5*^Ko^ male mice. (**B**) Micro-CT data were quantified (n=15–19). BMD, bone mineral density; BV/TV, bone volume relative to total tissue volume; Tb. V, trabecular volume; Tb. Th, trabecular bone thickness; Tb. N, trabecular bone number; Tb. Sp, trabecular bone space. (**C**) Quantitative analysis of the levels of receptor activator of nuclear factor kappa-B ligand (RANKL), osteoprotegerin (OPG), and bone morphogenic protein 2 (BMP2) in the sera from WT and *Prdx5*^Ko^ male mice at 12 weeks (n=4–8). (**D**) Representative tartrate-resistant acid phosphatase (TRAP) and alkaline phosphatase (ALP) staining images of the mouse femora. TRAP- or ALP-positive cells were stained as purple, and the bone was counterstained with Fast Green as blue. Scale bar, 100 μm. (**E**) Quantification of the TRAP- or ALP-positive cells shown in (**D**) (n=6–10). Graph depicts mean ± SEM. *p<0.05, **p <0.01 via an unpaired two-tailed Student's t-test compared to WT.

*Figure 3 continued on next page*

*Figure 3 continued*

The online version of this article includes the following source data and figure supplement(s) for figure 3:

**Figure supplement 1.** Female mice show normal phenotypes.

**Figure supplement 2.** Testosterone and androgen receptor (AR) expression levels are suppressed in *Prdx5*<sup>Ko</sup> male mice.

**Figure supplement 2—source data 1.** Full blots for *Figure 3—figure supplement 2B and C*.

**Figure supplement 2—source data 2.** Original blots for *Figure 3—figure supplement 2B and C*.

bone was trepanned, mice were treated with BMP2 or phosphate-buffered saline (PBS) for 3 weeks. Newly formed bones were observed in BMP2-administered mice; however, *Prdx5*<sup>Ko</sup> mice showed a lesser extent of new bone formation than WT (*Figure 4A*). Immunostaining analysis was performed to measure the cross-sectional area and bone volume. A larger puncture and smaller bone volume were observed in the calvaria of *Prdx5*<sup>Ko</sup> male mice than in those of WT (*Figure 4B*). The BMP2-restored lesions in *Prdx5*<sup>Ko</sup> male mice were thinner than those in WT mice. In contrast, the number of TRAP-positive cells were similar in *Prdx5*<sup>Ko</sup> and WT mice. However, *Prdx5*<sup>Ko</sup> mice had fewer ALP-positive osteoblasts than WT (*Figure 4C and D*). These results imply that *Prdx5* plays an essential role in osteoblast-mediated bone regeneration.

## Prdx5 co-localizes and interacts with hnRNPK in response to BMP2 stimulation

Prdx5 expression increased during osteoblast differentiation (*Figure 2*), which suggests that Prdx5 acts as a positive regulator of osteoblast differentiation. To understand the role of Prdx5 in osteoblasts, we investigated Prdx5-interacting proteins using LC–MS/MS after immunoprecipitation (IP) with a Prdx5 antibody using in vitro differentiated osteoblasts (*Figure 5A*). We identified 43 Prdx5-associated proteins (*Table 1*). Gene ontology (GO) analysis with these 43 proteins showed RNA splicing to be the only significant biological pathway, suggesting the involvement of Prdx5 through an RNA-related mechanism (*Figure 5B*). To determine the interacting proteins responsive to BMP2, we focused on BMP2-specific proteins. We classified 20 proteins as BMP2-specific interacting proteins (*Figure 5A* and *Table 2*). Because Prdx5 was localized in the nucleus after BMP2 stimulation (*Figure 5—figure supplement 1*) and to understand the function of Prdx5 in cell differentiation, we focused on nuclear proteins, hnRNPs. STRING analysis showed that hnRNPK was close to Prdx5 (*Figure 5C*). hnRNPK has been studied in osteoclasts and osteoblasts (*Fan et al., 2015*; *Stains et al., 2005*). Here, we confirmed the localization of Prdx5 and hnRNPK at the single-cell level (*Figure 6A*). After BMP2 stimulation, Prdx5 and hnRNPK were co-localized in the nucleus and cytosol in osteoblasts. We also confirmed the interaction between Prdx5 and hnRNPK using IP (*Figure 6B*). To clarify the relationship between Prdx5 and hnRNPK, we compared hnRNPK localization in *Prdx5*<sup>Ko</sup> cells after BMP2 stimulation. Interestingly, hnRNPK was localized only in the nucleus in *Prdx5*<sup>Ko</sup> cells, whereas it was observed both in the cytosol and nucleus in WT (*Figure 6C*). To verify the microscopic data, the hnRNPK levels were examined in the nuclear and cytoplasmic fractions of *Prdx5*<sup>Ko</sup> osteoblasts (*Figure 6D*). Higher levels of hnRNPK were detected in the nuclear fraction of *Prdx5*<sup>Ko</sup> osteoblasts than in that of the WT osteoblasts; the expression was similar in the absence of BMP2. These data suggest that Prdx5 may control the localization of hnRNPK in osteoblasts.

Previous studies have reported the role of hnRNPK as a repressor of *osteocalcin* transcription in osteoblasts (*Niger et al., 2011*; *Stains et al., 2005*). Because Prdx5 acted as an activator of osteoblast differentiation in our study, and *Bglap* levels were attenuated in osteoblasts from *Prdx5*<sup>Ko</sup> mice (*Figure 2C*), we assumed that Prdx5 inhibits hnRNPK to regulate osteocalcin. We performed a reporter assay using the *Bglap* promoter to verify whether Prdx5 affects *Bglap* expression (*Figure 6E*). We found that Prdx5 knockdown suppressed the *Bglap* activity that was rescued by Prdx5 overexpression. Further, to determine whether Prdx5 regulates hnRNPK accumulation on the *Bglap* promoter, we conducted chromatin immunoprecipitation (ChIP) assay on osteoblasts from WT and *Prdx5*<sup>Ko</sup> mice (*Figure 6—figure supplement 1*). In osteoblasts under BMP2 stimulation, hnRNPK accumulation on the *Bglap* promoter was much greatly suppressed than that of the pre-osteoblasts from the WT. However, in *Prdx5*-deficient osteoblasts, hnRNPK accumulation was highly increased under BMP2 stimulation compared to that in the WT. Altogether, these results indicate that Prdx5 interacts with

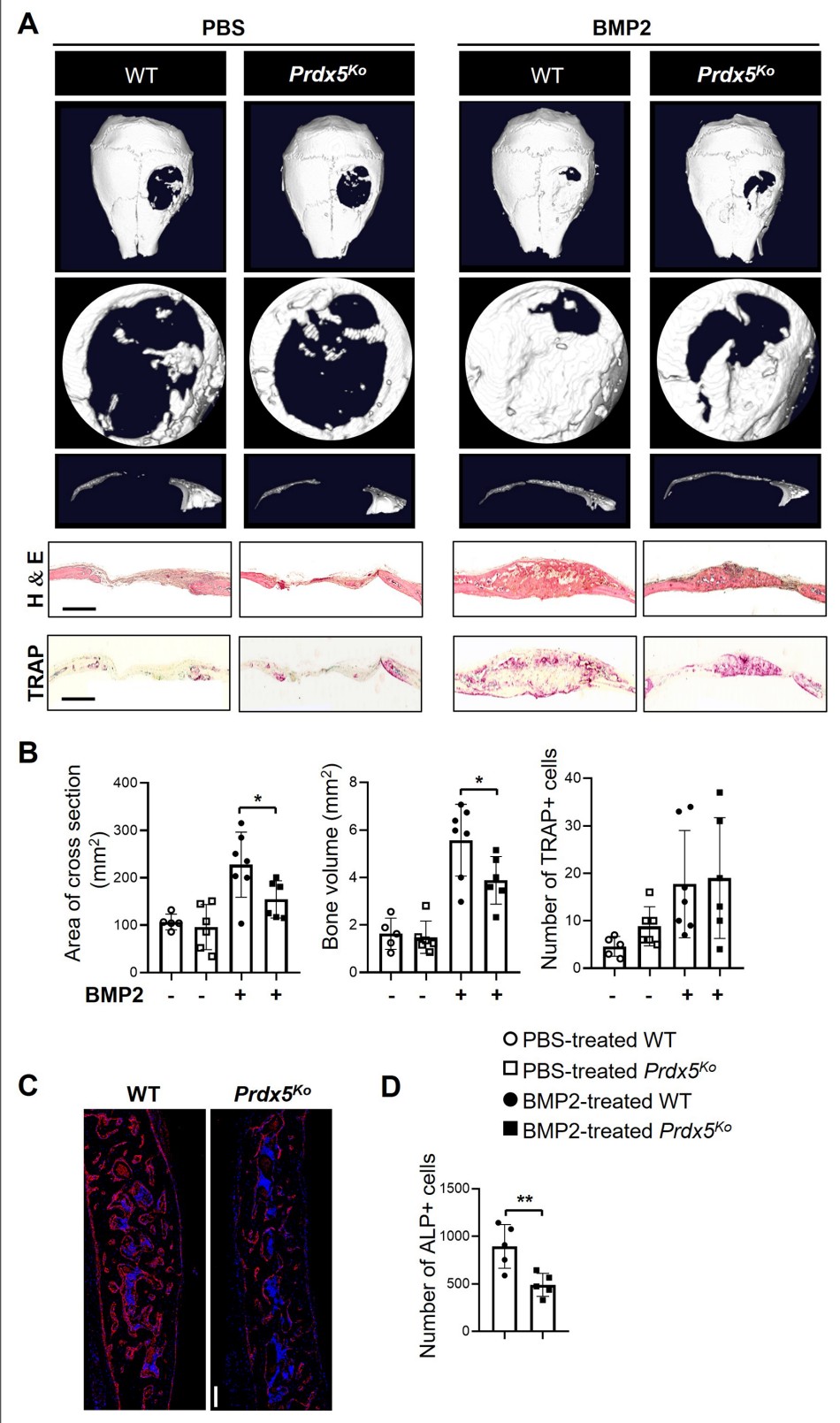

**Figure 4.** *Prdx5*[Ko] male mice show reduced bone healing after bone morphogenic protein 2 (BMP2) induction. (**A**) Representative micro-CT images of the calvarial defect model after 3 weeks of implantation with phosphate-buffered saline (PBS)- or BMP2-containing sponges. The representative images show various shapes: whole (top), the hole from each image (middle), and the cross-section (bottom) from each hole. Representative hematoxylin–

*Figure 4 continued on next page*

*Figure 4 continued*

eosin and tartrate-resistant acid phosphatase (TRAP) staining images of the calvarial bone section from each group. Scale bar, 1000 μM. (**B**) Measurement of the cross-sectional area, new bone formation, and number of TRAP-positive cells at the calvarial defect site (n=5–7). (**C**) Representative images of alkaline phosphatase (ALP) staining (scale bar, 100 μM) and (**D**) quantification of the number of ALP-positive cells of C (n=5). ALP-positive cells were stained red, while DAPI-positive cells were counterstained blue. Graph depicts mean ± SEM. *p<0.05, **p <0.01 via an unpaired two-tailed Student's t-test.

The online version of this article includes the following figure supplement(s) for figure 4:

**Figure supplement 1.** *Prdx5*^Ko^ male mice show reduced bone turnover.

hnRNPK in osteoblasts to transport hnRNPK from the nucleus to the cytoplasm, leading to osteocalcin activation to induce osteoblast differentiation.

## Expression of osteoclast-related genes increased in *Prdx5*^Ko^ osteoclasts

As *Prdx5*^Ko^ mice showed an increase in the number of TRAP-positive osteoclasts in the femurs (*Figure 3E*), and BMMs from *Prdx5*^Ko^ mice differentiated into more osteoclasts than those in WT (*Figure 2E*), we analyzed the transcriptome profiles of BMMs and osteoclasts through RNA-seq (*Figure 7*). The BMMs were isolated from WT and *Prdx5*^Ko^ mice, and the cells were differentiated into osteoclasts via RANKL stimulation for 4 days in vitro. The number of reads ranging from 72,748,470–86,717,526 was generated, and the trimmed clean reads were mapped to the mouse reference genome with 97–98% alignment rates (*Table 3*). BMMs and osteoclasts were clearly separated by principal component analysis (PCA) (*Figure 7A*). A comparison of differentially expressed genes (DEGs) between WT and *Prdx5*^Ko^ cells revealed 214 DEGs in BMMs, whereas 1257 genes were detected in osteoclasts (*Figure 7B and C*). Among the 214 genes, 61 (28.5%) were upregulated and 153 (71.5%) were downregulated in *Prdx5*^Ko^ mice compared to those in WT. However, approximately half of DEGs were up- and downregulated in *Prdx5*^Ko^ osteoclasts (51% and 49%, respectively). These results suggest that Prdx5 acts as an activator of gene expression in osteoclast precursors. However, the levels of these genes decrease during osteoclastogenesis. In GO analysis, the DEGs were found to be involved in the immune response (*Figure 7—figure supplement 1*).

We hypothesized that *Prdx5* deficiency results in a positive regulation of osteoclast differentiation. In GO analysis, the downregulated DEGs in *Prdx5*^Ko^ osteoclasts were involved in cell cycle regulation and cell division, while the upregulated DEGs were enriched through signaling and osteoclast differentiation (*Figure 7—figure supplement 1*). When we examined osteoclast-related genes, 25 out of 36 DEGs were upregulated in *Prdx5*^Ko^ osteoclasts (*Figure 7D*). Interestingly, the levels of transcription factors (*Nfatc1*, *Fos*, and *Irf9*) that regulated the early response of osteoclast differentiation were suppressed in *Prdx5*^Ko^ osteoclasts. In contrast, osteoclast maker genes (*Ocstamp*, *Calcr*, *Dcstamp*, *Itgb3*, and *Oscar*), which are highly expressed in mature osteoclasts, were upregulated in *Prdx5*^Ko^ osteoclasts.

## Discussion

Osteoporosisis is an excessive reduction in bone mass and is a major health issue in the elderly population (*Demontiero et al., 2012*). Clinically, some therapeutic treatments are available to induce osteoblast and reduce osteoclast activities (*Milat and Ebeling, 2016*). However, these treatments are associated with severe side effects, including heart issues, kidney damage, and osteonecrosis of the jaw (*Compston et al., 2019*; *Saag et al., 2017*). Therefore, a novel drug with curative and fewer side effects is urgently needed to treat osteoporosis.

Here, we assessed the critical functions of Prdx5 in bone homeostasis. Prdx5 expression increased during osteoblast differentiation and decreased during osteoclast differentiation. Genetically deficient *Prdx5* male mice developed osteoporosis-like phenotypes, suggesting that Prdx5 is important in bone remodeling. In osteoblasts, both Prdx5 and hnRNPK were co-localized in the nucleus and cytosol. Prdx5 regulated the hnRNPK-mediated osteocalcin transcription. In osteoclasts, Prdx5 acted as an inhibitor, as revealed by the upregulation of osteoclast-related genes in *Prdx5*^ko^ cells. We demonstrated that Prdx5 is a novel positive regulator of osteoblast differentiation, and that it also regulated

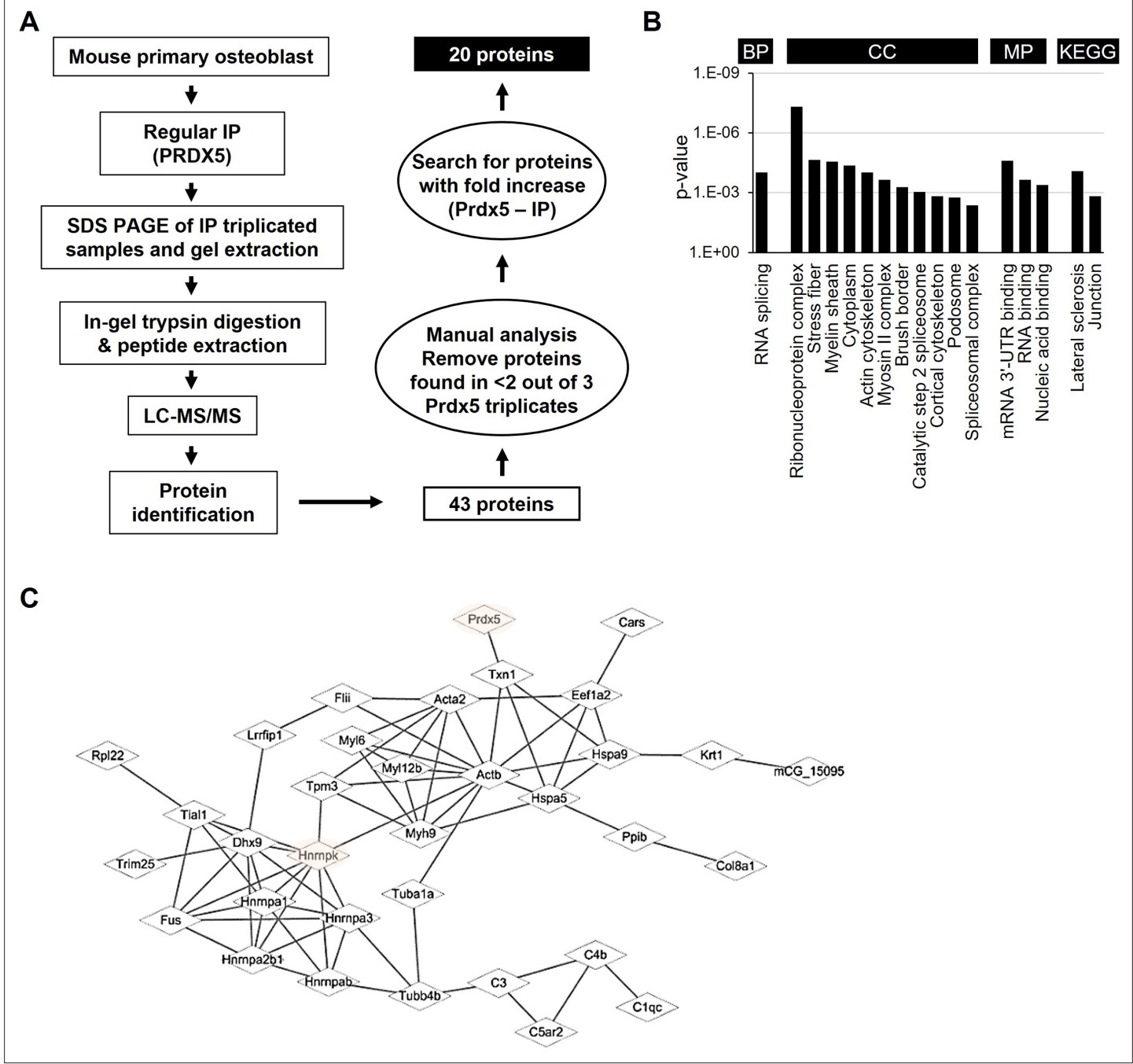

**Figure 5.** Identification of peroxiredoxin 5 (Prdx5)-interacting proteins during osteoblast differentiation. (**A**) Schematic representation of the experimental design of immunoprecipitation (IP) and liquid chromatography combined with tandem mass spectrometry (LC–MS/MS). Total 20 proteins were identified as potential candidates binding to Prdx5 in osteoclasts. (**B**) Gene ontology (GO) analysis results with 43 proteins are shown by biological process (BP), cellular component (CC), molecular function (MF), and Kyoto encyclopedia of genes and genomes (KEGG). (**C**) The interaction of Prdx5 with the 43 proteins identified in the MS/MS analysis was constructed using the STRING database.

The online version of this article includes the following source data and figure supplement(s) for figure 5:

**Figure supplement 1.** Bone morphogenic protein 2 (BMP2) induces nuclear translocation of peroxiredoxin 5 (Prdx5).

**Figure supplement 1—source data 1.** Full blots for *Figure 5—figure supplement 1A*.

**Figure supplement 1—source data 2.** Original blots for *Figure 5—figure supplement 1*.

**Table 1.** Forty-three peroxiredoxin 5 (Prdx5)-interacting proteins identified via liquid chromatography with tandem mass spectrometry (LC–MS/MS) analysis.

| Accessions | Description | Gene | Localization | Mol% | | | | | |
|---|---|---|---|---|---|---|---|---|---|
| | | | | CTRL_1 | CTRL_2 | CTRL_3 | BMP2_1 | BMP2_2 | BMP2_3 |
| P99029 | Peroxiredoxin-5, mitochondrial | *Prdx5* | Cytoplasm | 39.2 | 56.8 | 48.88 | 34.58 | 30.27 | 31.835 |
| Q6ZWQ9 | MCG5400 | *Myl12b* | Cytoplasm | | | | | 5.799 | 7.595 |
| P62737 | Actin, aortic smooth muscle | *Acta2* | Cytoplasm | | | | | 7.174 | 6.615 |
| P10639 | Thioredoxin | *Txn* | Cytoplasm | | 3.156 | 2.302 | 5.176 | 3.243 | 3.411 |
| Q8VDD5 | Myosin-9 | *Myh9* | Cytoplasm | | | | 3.059 | 2.31 | 3.204 |
| A0A075B5L7 | Immunoglobulin κ variable 4–80 (Fragment) | *Igkv4-80* | Other | 5.422 | 6.638 | | 2.118 | 2.998 | 3.152 |
| A0A0B4J1K5 | Immunoglobulin λ variable 3 (Fragment) | *Iglv3* | Other | 2.222 | 2.72 | 1.984 | 1.961 | 1.229 | 2.946 |
| O88569 | Heterogeneous nuclear ribonucleoproteins A2/B1 | *Hnrnpa2b1* | Nucleus | | 0.979 | | | 2.604 | 2.739 |
| Q8CFQ9 | Fusion, derived from t(1216) malignant liposarcoma | *Fus* | Nucleus | 0.533 | 0.653 | 0.476 | | | 1.809 |
| P04104 | Keratin, type II cytoskeletal 1 | *Krt1* | Cytoplasm | 3.556 | 2.938 | 2.143 | 1.647 | 1.327 | 1.395 |
| E9Q1Z0 | Keratin 90 | *Krt90* | Cytoplasm | 2.133 | | 2.46 | | 1.523 | 1.24 |
| P67984 | 60S ribosomal protein L22 | *Rpl22* | Cytoplasm | | | | 1.804 | | 1.189 |
| A0A1W2P6G5 | Myosin light polypeptide 6 | *Myl6* | Cytoplasm | | | | 3.765 | 1.081 | 1.137 |
| Q9JJ28 | Protein flightless-1 homolog | *Flii* | Nucleus | 0.622 | 0.762 | 1.111 | 1.098 | 0.688 | 1.137 |
| P68369 | Tubulin α-1A chain | *Tuba1a* | Cytoplasm | | | | 2.902 | 1.032 | 1.085 |
| P21107 | Tropomyosin α-3 chain | *Tpm3* | Cytoplasm | | | | | 1.032 | 1.085 |
| Q20BD0 | Heterogeneous nuclear ribonucleoprotein A/B | *Hnrnpab* | Nucleus | 0.8 | 2.607 | 0.714 | | 0.934 | 0.982 |
| Q5EBP8 | Heterogeneous nuclear ribonucleoprotein A1 | *Hnrnpa1* | Nucleus | | | | | 0.885 | 0.93 |
| P70318 | Nucleolysin TIAR | *Tial1* | Nucleus | 0.711 | 0.871 | | | 1.179 | 0.827 |
| B2M1R6 | Heterogeneous nuclear ribonucleoprotein K | *Hnrnpk* | Nucleus | | | | | 1.081 | 0.724 |
| P20029 | Endoplasmic reticulum chaperone BiP | *Hspa5* | Cytoplasm | 0.444 | | 0.714 | | 0.688 | 0.724 |
| P38647 | Stress-70 protein, mitochondrial | *Hspa9* | Cytoplasm | 2.133 | | 0.714 | | 0.442 | 0.724 |
| P68372 | Tubulin β-4B chain | *Tubb4b* | Cytoplasm | | | | 1.098 | | 0.724 |
| Q9CPN9 | Complement C1q subcomponent subunit C | *C1qc* | Extracellular space | 1.156 | 1.415 | 1.032 | | 0.639 | 0.672 |
| Q02105 | RIKEN cDNA 2210010C04 gene | *2210010C04Rik* | Extracellular space | 1.067 | 1.306 | 0.952 | 0.941 | 0.59 | 0.62 |

*Table 1 continued on next page*

*Table 1 continued*

| Accessions | Description | Gene | Localization | Mol% CTRL_1 | CTRL_2 | CTRL_3 | BMP2_1 | BMP2_2 | BMP2_3 |
|---|---|---|---|---|---|---|---|---|---|
| A0A087WNU6 | Leucine-rich repeat flightless-interacting protein 1 (Fragment) | *Lrrfip1* | Cytoplasm | | | | 0.392 | 0.491 | 0.517 |
| Q8BG05-2 | Isoform 2 of heterogeneous nuclear ribonucleoprotein A3 | *Hnrnpa3* | Nucleus | | | | | 0.934 | 0.465 |
| A0A1D5RLD8 | Glyceraldehyde-3-phosphate dehydrogenase | *GM10358* | Other | | | | | 0.442 | 0.465 |
| P62631 | Elongation factor 1-α2 | *Eef1a2* | Cytoplasm | | | | | 0.295 | 0.31 |
| P01029 | Complement C4-B | *C4a/C4b* | Extracellular space | 0.267 | 0.326 | 0.397 | 0.235 | 0.147 | 0.258 |
| F7DBB3 | AHNAK nucleoprotein 2 (Fragment) | *Ahnak2* | Cytoplasm | 0.178 | 0.435 | 0.476 | | 0.393 | 0.207 |
| Q00780 | Collagen α-1(VIII) chain | *Col8a1* | Extracellular Space | | | 0.714 | | 0.197 | 0.207 |
| Q61510 | E3 ubiquitin/ISG15 ligase TRIM25 | *Trim25* | Cytoplasm | | | | | | |
| P01027 | Complement C3 | *C3* | Extracellular Space | 0.622 | 0.544 | 0.397 | | 0.098 | 0.207 |
| P60710 | Actin, cytoplasmic 1 | *Actb* | Cytoplasm | | | | 14.745 | 8.206 | |
| Q9Z1R9 | MCG124046 | *Prss1* (includes others) | Extracellular Space | 2.311 | | 2.063 | | 1.278 | |
| H3BJS5 | Melanoma inhibitory activity protein 2 (Fragment) | *Mia2* | Cytoplasm | | | | 0.863 | 0.541 | |
| O70133 | ATP-dependent RNA helicase A | *Dhx9* | Nucleus | | | | 0.157 | 0.197 | |
| P24369 | Peptidyl-prolyl *cis-trans* isomerase B | *Ppib* | Cytoplasm | 13.511 | 10.12 | 12.063 | | | |
| F6T9C3 | Translation initiation factor eIF-2B subunit ε (Fragment) | *Eif2b5* | Cytoplasm | | 1.959 | 1.429 | | | |
| Q8QZT1 | Acetyl-CoA acetyltransferase, mitochondrial | *Acat1* | Cytoplasm | 1.333 | | 0.556 | | | |
| Q9ER72 | Cysteine-tRNA ligase, cytoplasmic | *Cars* | Cytoplasm | 0.267 | | 0.238 | | | |
| P56480 | ATP synthase subunit β, mitochondrial | *Atp5f1b* | Cytoplasm | 1.067 | 1.306 | | | | |

osteoclastogenesis. Our study indicated the beneficial pharmacological effect of Prdx5 in the maintenance of bone mass during the formation of skeletal tissues.

Six members of the Prdx family reportedly exhibit antioxidant activities owing to the presence of CXXC amino acid sequences (*Chae et al., 1994*; *Rhee et al., 2001*). In mammals, Prdx5 is a unique member of the atypical 2-Cys subfamily and is expressed ubiquitously in all tissues (*Rhee et al., 2001*). It is present in the cytosol, peroxisomes, and mitochondria (*Rhee et al., 2012*). *Prdx5* deficiency increases the susceptibility to high-fat diet-induced obesity and metabolic abnormalities (*Kim et al., 2018*; *Kim et al., 2018*). In this study, we first investigated the changes in osteogenesis or bone mass formation by Prdx5 and then confirmed the role of Prdx5 in osteogenic processes. The *Prdx5*^Ko male

**Table 2.** Peroxiredoxin 5 (Prdx5)-interacting proteins detected only in the bone morphogenic protein 2 (BMP2)-treated group.

| Accession | Description | Gene | Localization | Avg of mol% |
|---|---|---|---|---|
| Q8VDD5 | Myosin-9 | Myh9 | Cytoplasm | 2.858 |
| A0A1W2P6G5 | Myosin light polypeptide 6 | Myl6 | Cytoplasm | 1.994 |
| P68369 | Tubulin α-1A chain | Tuba1a | Cytoplasm | 1.673 |
| A0A087WNU6 | Leucine-rich repeat flightless-interacting protein 1 (Fragment) | Lrrfip1 | Cytoplasm | 0.467 |
| P60710 | Actin, cytoplasmic 1 | Actb | Cytoplasm | 11.48 |
| P62737 | Actin, aortic smooth muscle | Acta2 | Cytoplasm | 6.895 |
| Q6ZWQ9 | MCG5400 | Myl12b | Cytoplasm | 6.879 |
| O88569 | Heterogeneous nuclear ribonucleoproteins A2/B1 | Hnrnpa2b1 | Nucleus | 2.672 |
| P67984 | 60S ribosomal protein L22 | Rpl22 | Cytoplasm | 1.496 |
| P21107 | Tropomyosin α-3 chain | Tpm3 | Cytoplasm | 1.059 |
| P68372 | Tubulin β-4B chain | Tubb4b | Cytoplasm | 0.911 |
| Q5EBP8 | Heterogeneous nuclear ribonucleoprotein A1 | Hnrnpa1 | Nucleus | 0.907 |
| B2M1R6 | Heterogeneous nuclear ribonucleoprotein K | Hnrnpk | Nucleus | 0.902 |
| H3BJS5 | Melanoma inhibitory activity protein 2 (Fragment) | Mia2 | Cytoplasm | 0.702 |
| Q8BG05-2 | Isoform 2 of heterogeneous nuclear ribonucleoprotein A3 | Hnrnpa3 | Nucleus | 0.699 |
| A0A1D5RLD8 | Glyceraldehyde-3-phosphate dehydrogenase | GM10358 | Other | 0.454 |
| P62631 | Elongation factor 1-α2 | Eef1a2 | Cytoplasm | 0.302 |
| Q00780 | Collagen α-1(VIII) chain | Col8a1 | Extracellular space | 0.202 |
| Q61510 | E3 ubiquitin/ISG15 ligase TRIM25 | Trim25 | Cytoplasm | 0.202 |
| O70133 | ATP-dependent RNA helicase A | Dhx9 | Nucleus | 0.177 |

mice showed a significant reduction in bone mass, presenting 15% lower BMD and 10% less trabecular volume compared to that obtained with WT. Hence, it can be suggested that Prdx5 affects the bone turnover. *Prdx5* deficiency markedly inhibited osteoblast differentiation and increased osteoclast differentiation in vitro. Indeed, the bone healing rate and osteocyte population decreased in *Prdx5*<sup>Ko</sup> male mice. Interestingly, Prdx5 may interact with hnRNPK in osteoblasts. Given the reduced bone mass in *Prdx5*<sup>Ko</sup> mice, we investigated the function of Prdx5 in osteoclasts. However, we did not focus on the role of Prdx5 in osteoclasts, because its expression was extremely low after RANKL stimulation. Our results imply that Prdx5 primarily acts in osteoblasts, and it may not be necessary for osteoclasts.

In our studies, osteoporotic phenotypes were observed only in *Prdx5*<sup>Ko</sup> male mice. Testosterone and AR are required for maintaining bone mass in male mice (**Wu et al., 2019**). Therefore, we compared the testosterone levels in the serum from *Prdx5*<sup>Ko</sup> and WT mice (**Figure 3—figure supplement 2**). Interestingly, the levels of testosterone and their receptor, AR, were significantly decreased in *Prdx5*<sup>Ko</sup>. It should be noted that *Prdx5*<sup>Ko</sup> mice were fertile gross normal, although their testosterone levels were lower than that of WT. hnRNPK inhibits the translation of AR mRNA in prostate cancer (**Mukhopadhyay et al., 2009**). It is possible that increased hnRNPK in osteoblasts from *Prdx5*<sup>Ko</sup> can reduce AR expression levels, leading to osteoporosis in *Prdx5*<sup>Ko</sup> mice.

To determine the antioxidative role of Prdx5 in bone cell differentiation, we determined the ROS levels in BMP2-treated osteoblasts. ROS are involved in bone remodeling. They accelerate the resorption of the mineralized matrix and inhibit osteoblast differentiation (**Ren et al., 2021**). Human bone marrow cells actively differentiate into osteoclasts when hydrogen peroxide was added (**Baek et al., 2010**). Indeed, ROS inhibits osteoblast differentiation by blocking osteogenic signaling pathways (**Bai et al., 2004**). Cytokine stimulation, such as RANKL and BMP2, and their signaling cascades, including TRAF6, Smads 1/5/8, Rac1, and Noxs, elicit ROS production during bone cell differentiation (**Kanzaki et al., 2013**; **Mandal et al., 2011**; **Pan et al., 2022**). Nox4, the predominant isotype of

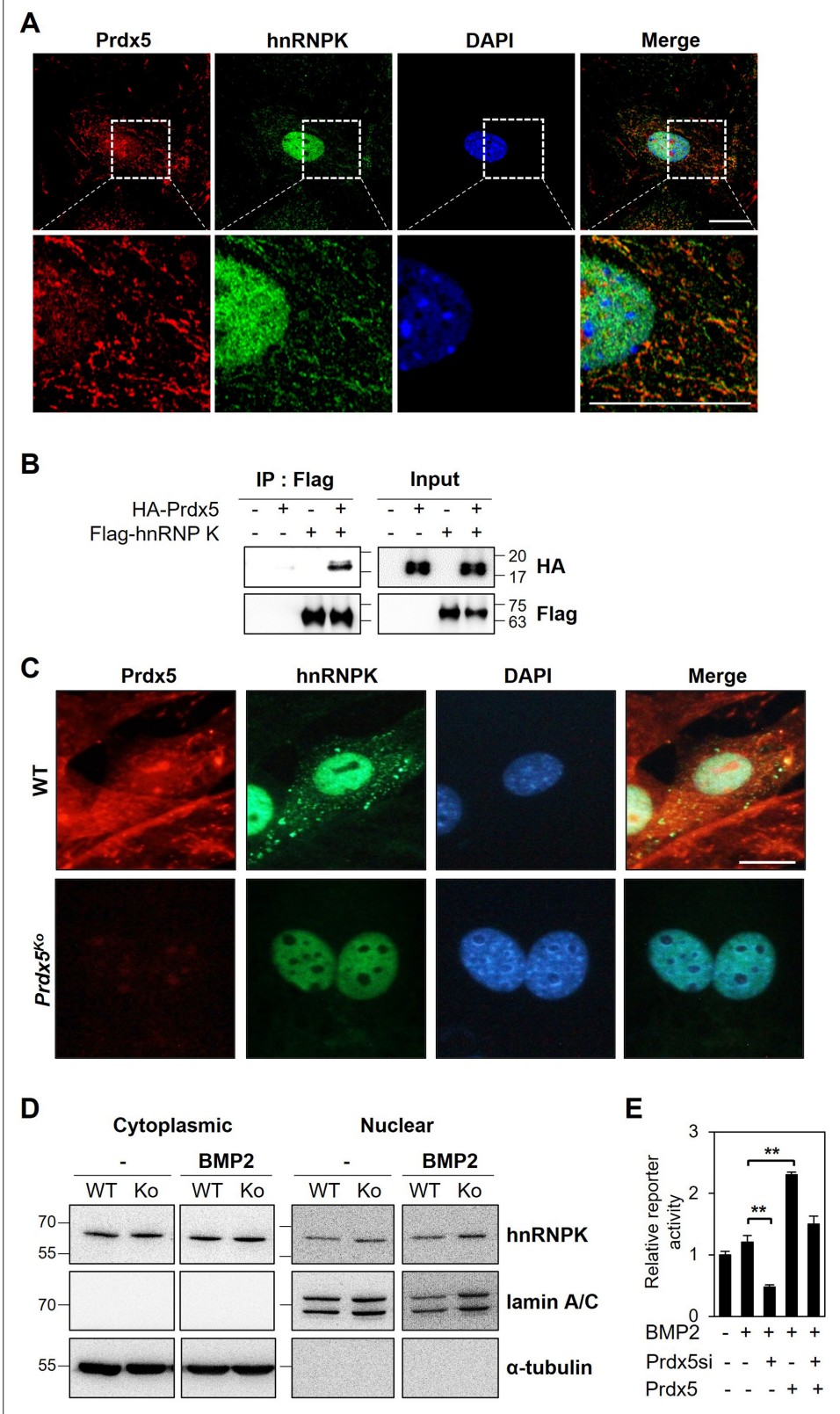

**Figure 6.** Heterogeneous nuclear ribonucleoprotein K (hnRNPK) interacts with peroxiredoxin 5 (Prdx5) in osteoblasts. (**A**) To determine co-localization, osteoblasts were stained with antibodies against Prdx5 and hnRNPK, and images were acquired via confocal microscopy (scale bar, 20 μm). The upper images were magnified as depicted by the dotted box in the lower images. (**B**) Immunoprecipitation (IP) was performed using HEK293T

*Figure 6 continued on next page*

*Figure 6 continued*

cells expressing various combinations of HA-tagged Prdx5 and flag-tagged hnRNPK. (**C**) Osteoblasts were differentiated from the precursors derived from wild-type (WT) and *Prdx5*[Ko] mice via bone morphogenic protein 2 (BMP2) treatment for 7 days. hnRNPK localization was analyzed via confocal microscopy (scale bar, 20 µm). (**D**) hnRNPK levels were determined in the cytoplasmic and nuclear fractions of WT and *Prdx5*[Ko] cells. Osteoblasts were harvested on day 7 after BMP2 stimulation. (**E**) Osteocalcin luciferase assay was performed using MC3T3-E1 cells differentially expressing Prdx5 and BMP2 stimulation. Data are presented as mean ± SD. **p <0.01 via an unpaired two-tailed Student's t-test.

The online version of this article includes the following source data and figure supplement(s) for figure 6:

**Source data 1.** Full blots for *Figure 6B and D*.

**Source data 2.** Original blots for *Figure 6B and D*.

**Figure supplement 1.** Heterogeneous nuclear ribonucleoprotein K (hnRNPK) accumulation is enhanced in *Prdx5*[Ko] osteoblasts.

Nox in osteoblasts, is required for BMP2-stimulated ROS generation during osteoblast differentiation (*Mandal et al., 2011*). BMP2 stimulation did not alter ROS production in *Prdx5*-deficient cells. Although Nox subunits were downregulated in pre-osteoblasts lacking Prdx5, *Nox4* and *Rac1* expression levels were not significantly altered between *Prdx5*[Ko] and WT. We found that Prdx5 is involved in ROS generation during osteoblast differentiation, which is necessary for BMP2-mediated ROS production. However, this mechanism is an early response in the cytoplasm, and ROS are converted to water and oxygen by glutathione peroxidases, catalases, and Prdxs, which should be suppressed during osteogenic differentiation (*Atashi et al., 2015*). Therefore, we suggest that other antioxidants in the cytoplasm or mitochondria can compensate the enzymatic effect of Prdx5 that were induced during osteoblast or osteoclast differentiation, such as Prdx2 and Prdx3 (*Figure 1*). Further studies are required to elucidate the relationship between ROS and Prdx5 in bone cells, particularly, in terms of mitochondrial functions. In this study, we primarily focused on the role of Prdx5 in the nucleus.

hnRNPs are a family of nuclear proteins that function in mRNA biogenesis, including pre-mRNA splicing (*Expert-Bezançon et al., 2002*), transport of mRNA from the nucleus to the cytosol (*Michael et al., 1997*), and translation (*Ostareck et al., 1997*). hnRNPK is a unique member of this family, as it preferentially binds single-stranded DNA, whereas other hnRNPs bind RNA (*Siomi et al., 1994*). hnRNPK is a multifunctional molecule, which can act both in the cytosol and in the nucleus (*Krecic and Swanson, 1999*; *Mikula et al., 2010*). It has been implicated in various cellular processes, including gene transcription (*Michelotti et al., 1996*; *Tomonaga and Levens, 1996*) and chromatin remodeling (*Denisenko and Bomsztyk, 2002*), as well as in more typical functions of splicing and mRNA transport to the cytoplasm (*Dreyfuss et al., 1993*). *HNRNPK* mutation in humans causes a Kabuki-like

**Table 3.** Statistics of RNA-sequencing (RNA-seq) analysis.

| Original | Number of reads (sum of pairs) | After trimmed reads | Alignment rate (%) |
|---|---|---|---|
| Wt_BMMs-1_Read_Count | 79,081,594 | 77,290,554 | 97.74% |
| Wt_BMMs-2_Read_Count | 73,952,744 | 72,383,444 | 97.88% |
| Wt_BMMs-3_Read_Count | 84,004,640 | 81,523,436 | 97.05% |
| Wt_OCs-1_Read_Count | 75,068,812 | 73,463,268 | 97.86% |
| Wt_OCs-2_Read_Count | 79,729,144 | 77,767,782 | 97.54% |
| Wt_OCs-3_Read_Count | 81,017,248 | 79,417,460 | 98.03% |
| KO_BMMs-1_Read_Count | 77,677,924 | 75,695,744 | 97.45% |
| KO_BMMs-2_Read_Count | 76,958,152 | 75,180,826 | 97.69% |
| KO_BMMs-3_Read_Count | 73,431,784 | 71,442,572 | 97.29% |
| KO_OCs-1_Read_Count | 86,717,526 | 84,541,376 | 97.49% |
| KO_OCs-2_Read_Count | 81,819,488 | 79,737,322 | 97.46% |
| KO_OCs-3_Read_Count | 72,748,470 | 70,913,232 | 97.48% |

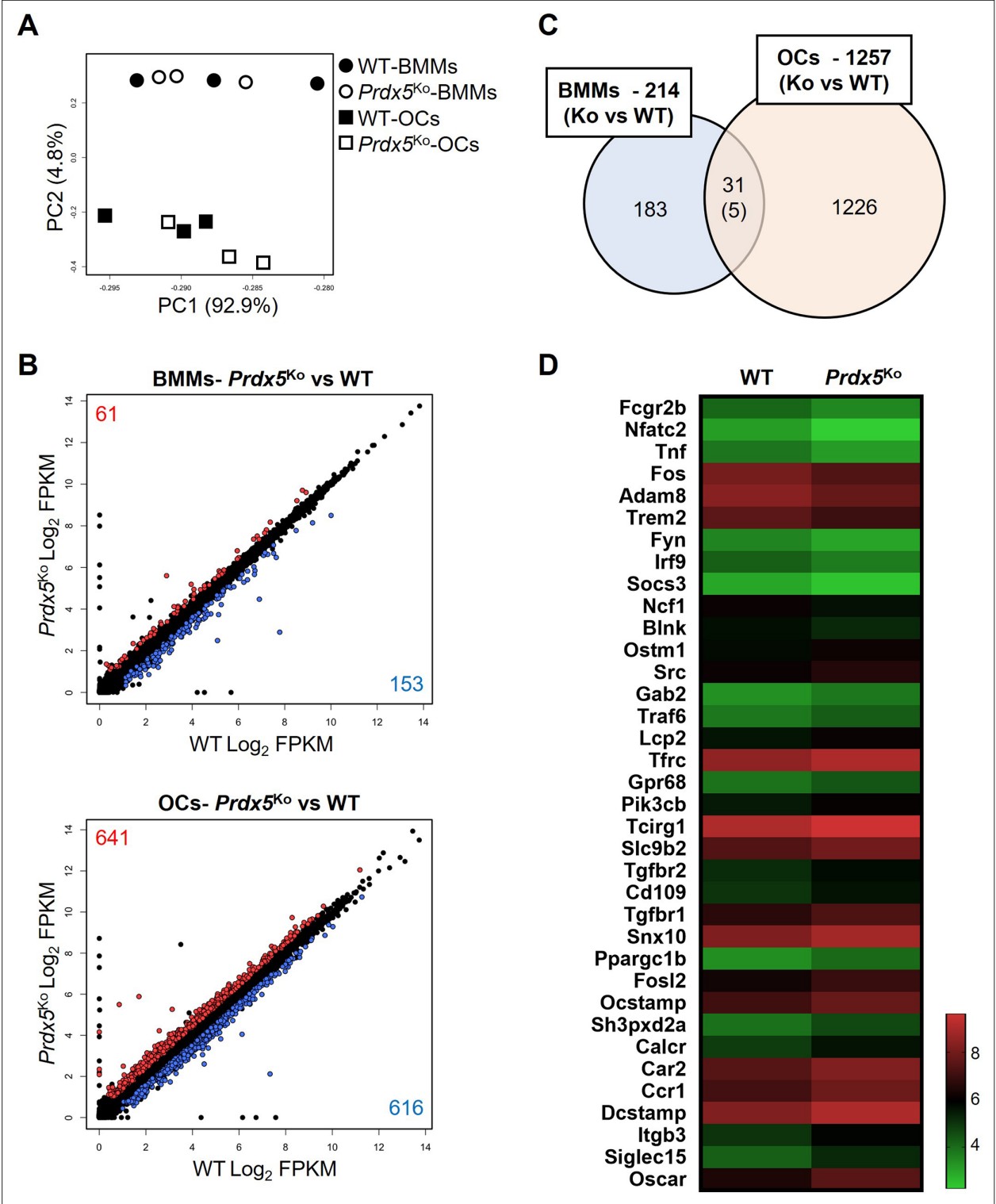

**Figure 7.** Osteoclast-related genes are highly expressed in peroxiredoxin 5 (*Prdx5*)-deficient osteoclasts. (**A**) Principal component analysis (PCA) of bone marrow-derived macrophages (BMMs) and osteoclasts (OCs) from wild-type (WT) and *Prdx5*[Ko] cells. Each circle or square represents the expression profile of one sample (n=3). (**B**) Differentially expressed genes (DEGs) in BMMs and OCs by comparison of *Prdx5*[Ko] versus WT are displayed on a scatter plot. Each dot indicates a single gene. Significantly upregulated DEGs in *Prdx5*[Ko] are indicated in red, while downregulated DEGs are indicated in blue (fragment per kilobase of exon per million fragments mapped [FPKM] >1, q-value <0.05). (**C**) Venn diagram indicates DEGs in BMMs and OCs. A total of 31 DEGs are overlapped in BMMs and OCs, and only five genes show opposite patterns, which are downregulated in *Prdx5*[Ko] OCs but upregulated in *Prdx5*[Ko] BMMs. (**D**) Heatmap analysis shows osteoclast-related DEGs. The z-score represents log$_2$ FPKM.

*Figure 7 continued on next page*

*Figure 7 continued*

The online version of this article includes the following figure supplement(s) for figure 7:

**Figure supplement 1.** Gene ontology (GO) analysis of differentially expressed genes (DEGs) via RNA-sequencing (RNA-seq) analysis.

syndrome with skeletal abnormalities and facial dysmorphism (*Wang et al., 2020*); patients with acute myeloid leukemia show aberrant hnRNPK expression (*Gallardo et al., 2015*). *Hnrnpk* deletion in mice is embryonically lethal, and haploinsufficiency results in developmental defects with skeletal disorders and post-natal death (*Au et al., 2018*; *Dentici et al., 2018*; *Gallardo et al., 2015*). hnRNPK acts as a transcription factor and regulates translation by binding to promoters. In cancer, hnRNPK binds to the promoter regions of *MYC* (MYC proto-oncogene) and *SRC* (SRC proto-oncogene) to elevate their transcription or binds to their mRNAs to control translation (*Naarmann et al., 2008*; *Perrotti and Neviani, 2007*; *Ritchie et al., 2003*). In amyotrophic lateral sclerosis, hnRNPK binds to antioxidant *NFE2L2* (NFE2-like BZIP transcription factor 2) and *GPX1* (glutathione peroxidase 1) transcripts (*Moujalled et al., 2017*). In the bone, it interacts with glycogen synthase kinase 3 beta to promote osteoclast differentiation (*Fan et al., 2015*). During osteoblast differentiation, hnRNPK binds to the promoter region of osteocalcin (*Bglap*) and represses its transcription (*Stains et al., 2005*).

Prdx5 was also expressed in the cytosol and nucleus (*Figure 7*). We examined Prdx5 translocation to the nucleus upon BMP2 induction. Our results suggested potential mechanisms through which transcriptional repression by hnRNPK may occur. The most likely scenario is that hnRNPK competitively binds to an unknown transcription factor (complex II) that binds to the putative CT-rich region of the *Bglap* promoter, resulting in the loss of an activator from the promoter and a net repression of gene transcription. Our results indicated that Prdx5 disturbed the binding potential of hnRNPK to suppress *Bglap* expression through an interaction between Prdx5 and hnRNPK and their translocation (*Figure 8*). We suggested that Prdx5 acts as an inducer of *Bglap* transcription by removing hnRNPK from the *Bglap* promoter. hnRNPK interacts with glycogen synthase kinase 3 beta during osteoclast differentiation via nuclear–cytoplasmic translocation (*Fan et al., 2015*). Further studies are needed to demonstrate the correlation between Prdx5 attenuation and hnRNPK translocation during osteoclastogenesis.

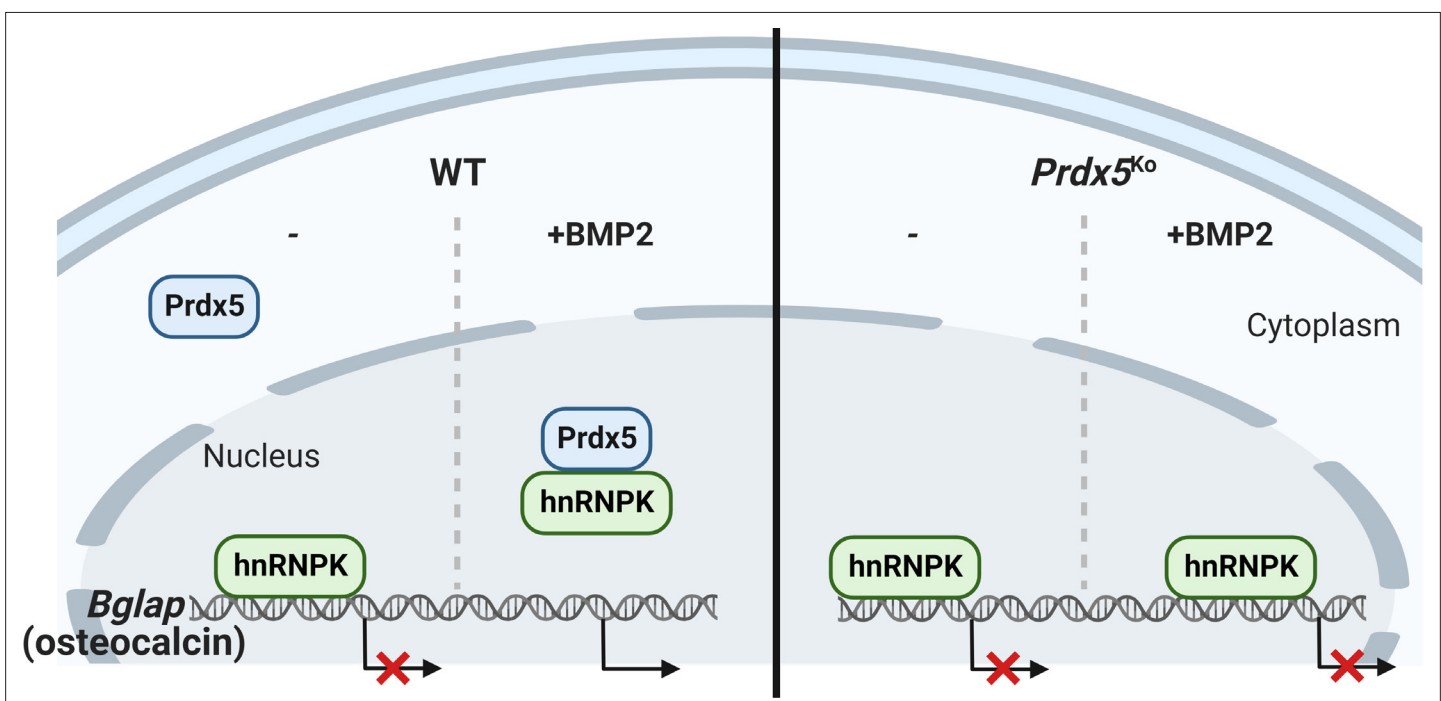

**Figure 8.** Heterogeneous nuclear ribonucleoprotein K (hnRNPK) accumulates on the *Bglap* promoter and inhibits its expression in the nucleus without bone morphogenic protein 2 (BMP2) stimulation. When BMP2 induces peroxiredoxin 5 (Prdx5) translocation into the nucleus, Prdx5 removes hnRNPK from the *Bglap* promoter by interacting with the latter, followed by osteocalcin transcription.

In conclusion, we identified a new mechanism through which Prdx5 regulates the hnRNPK–osteo-calcin axis in osteoblasts. Our study also indicates that Prdx5 controls osteoclast differentiation, which is mediated by osteoblast differentiation or the early stages of osteoclastogenesis. Therefore, Prdx5 is critical for bone remodeling.

# Materials and methods

**Key resources table**

| Reagent type (species) or resource | Designation | Source or reference | Identifiers | Additional information |
|---|---|---|---|---|
| Genetic Reagent (*Mus musculus*) | B6-*Prdx5*^tm1/J (*Prdx5*^Ko) | **Kim et al., 2018** | | |
| Cell line (*Mus musculus*) | MC3T3-E1 | **Cho et al., 2018** | | Cell line was verified by KCLB and tested negative for mycoplasma. |
| Cell line (*Homo sapiens*) | HEK293T | ATCC | CRL-3216 | Cell line was STR profiled by KCLB and tested negative for mycoplasma. |
| Transfected construct (*Mus musculus*) | *Prdx5* siRNA | Genolution | | 5'-GCGUUAAUGACGUCUUUGUUU-3' 5'-ACAAAGACGUCAUUAACGCUU-3' |
| Recombinant DNA reagent | p3xFLAG-CMV-10 (plasmid) | Sigma-Aldrich | Cat#E7658 | Subcloned mouse hnRNPK (EcoR I-Kpn I) |
| Recombinant DNA reagent | pCMV-HA (plasmid) | Clontech Laboratories | Cat#635690 | Subcloned mouse Prdx5 (EcoR I-Bgl II) |
| Peptide, recombinant protein | BMP2 | Sino Biological | Cat#10426 | |
| Peptide, recombinant protein | Recombinant murine sRANKL | PeproTech | Cat#315–11 | |
| Peptide, recombinant protein | Recombinant murine M-CSF | PeproTech | Cat#315–02 | |
| Chemical compound, drug | Type I collagenase | Gibco | Cat#17018 | |
| Chemical compound, drug | Dispase | Roche | Cat#4942078001 | |
| Chemical compound, drug | BCIP/NBT alkaline phosphatase kit | Sigma-Aldrich | Cat#B1911 | |
| Chemical compound, drug | CM-H2DCFDA | Thermo Fisher Scientific | Cat#MP36103 | |
| Chemical compound, drug | Calcein | Sigma-Aldrich | Cat#C0875 | |
| Chemical compound, drug | Alizarin Red S | Sigma-Aldrich | Cat#A5533 | |
| Chemical compound, drug | Lipofectamine 3000 | Invitrogen | Cat#L3000001 | |
| Chemical compound, drug | Hematoxylin | Sigma-Aldrich | Cat#H9627 | |
| Chemical compound, drug | Bouin's solution | Sigma-Aldrich | Cat#HT10132 | |
| Chemical compound, drug | Acid Fuchsin | Sigma-Aldrich | Cat#F-8129 | |
| Chemical compound, drug | Aniline blue | Junsei | Cat#11466 | |
| Chemical compound, drug | Phosphomolybdic acid hydrate | Junsei | Cat#84235 | |

*Continued on next page*

*Continued*

| Reagent type (species) or resource | Designation | Source or reference | Identifiers | Additional information |
|---|---|---|---|---|
| Chemical compound, drug | Phosphotungstic acid hydrate | Junsei | Cat#84220 | |
| Chemical compound, drug | Fast Green FCF | Sigma-Aldrich | Cat#7252 | |
| Chemical compound, drug | FuGENE HD | Promega | Cat#E2311 | |
| Chemical compound, drug | RIPA lysis buffer | Thermo Fisher Scientific | Cat#89900 | |
| Chemical compound, drug | TRIzol reagent | Thermo Fisher Scientific | Cat#15596026 | |
| Chemical compound, drug | Trypsin Gold | Promega | Cat#V5280 | |
| Commercial assay or kit | TRAP Staining Kit | Cosmo Bio Co. | Cat#PMC-AK04-COS | |
| Commercial assay or kit | TRACP & ALP assay kit | TaKaRa | Cat#MF301 | |
| Commercial assay or kit | SYBR Green Master mix | Thermo Fisher Scientific | Cat#A25778 | |
| Commercial assay or kit | Reverse transcription kit | Thermo Fisher Scientific | Cat#18064022 | |
| Commercial assay or kit | Mouse RANKL ELISA | Abcam | Cat#ag100749 | |
| Commercial assay or kit | Mouse OPG ELISA | R&D systems | Cat#MOP00 | |
| Commercial assay or kit | Mouse BMP2 ELISA | LSBio | Cat#LS-F36595 | |
| Commercial assay or kit | Nuclear and cytoplasmic extraction kit | Thermo Fisher Scientific | Cat#78833 | |
| Commercial assay or kit | Luciferase assay system | Promega | Cat#E1500 | |
| Commercial assay or kit | RNeasy mini kit | Qiagen | Cat#74004 | |
| Commercial assay or kit | Testosterone ELISA | R&D Systems | Cat#KGE010 | |
| Commercial assay or kit | ChIP assay | Cell Signaling Technology | Cat#9003 | |
| Antibody | Anti-Prdx5 (mouse monoclonal) | Invitrogen | Cat#LF-MA0002 | (1:1000) |
| Antibody | Anti-Prdx5 (rabbit polyclonal) | Ab Frontier | Cat#LF-PA0010 | (1:500) |
| Antibody | Anti-hnRNPK (rabbit polyclonal) | Cell Signaling Technology | Cat#9081 | (1:1000) |
| Antibody | Anti-beta actin (mouse monoclonal) | Sigma | Cat#A5441 | (1:1000) |
| Antibody | Anti-ALP (rabbit polyclonal) | Abcam | Cat#ab229126 | (1:200) |
| Antibody | Anti-HA-Tag (mouse monoclonal) | Santa Cruz | Cat#sc-7392 | (1:1000) |
| Antibody | Anti-Flag (OctA)-probe (mouse monoclonal) | Santa Cruz | Cat#sc-166355 | (1:1000) |

*Continued on next page*

*Continued*

| Reagent type (species) or resource | Designation | Source or reference | Identifiers | Additional information |
|---|---|---|---|---|
| Antibody | Anti-lamin A/C (rabbit polyclonal) | Cell Signaling Technology | Cat#2032 | (1:1000) |
| Antibody | Anti-lamin B (rabbit polyclonal) | Ab Frontier | Cat#LF-PA50043 | (1:1000) |
| Antibody | Anti-tubulin alpha (mouse monoclonal) | Novus | Cat#NB100 | (1:1000) |
| Antibody | Anti-rabbit Alexa Fluor 488 (goat polyclonal) | Thermo Fisher Scientific | Cat#A-32731 | (1:200) |
| Antibody | Anti-mouse Alexa Fluor 555 (rabbit polyclonal) | Thermo Fisher Scientific | Cat#A-21427 | (1:200) |
| Antibody | Anti-androgen receptor (mouse monoclonal) | Santa Cruz | Cat#sc-7305 | (1:1000) |
| Antibody | Anti-Prdx1 (rabbit polyclonal) | Invitrogen | Cat#PA3-750 | (1:1000) |
| Antibody | Anti-Prdx2 (rabbit polyclonal) | Ab Frontier | Cat#LF-PA0007 | (1:500) |
| Antibody | Anti-Prdx3 (rabbit polyclonal) | Ab Frontier | Cat#LF-MA0329 | (1:500) |
| Antibody | Anti-Prdx4 (rabbit polyclonal) | Abcam | Cat#ab184167 | (1:500) |
| Antibody | Anti-Prdx6 (rabbit polyclonal) | Invitrogen | Cat#PA5-30320 | (1:1000) |
| Software, algorithm | GraphPad Prism software 8 | https://graphpad.com | | |
| Software, algorithm | ImageJ software | https://imagej.nhi.gov/ij | | |

## Animal experiments

All animals were housed in a specific pathogen-free facility following the guidelines provided in the Guide for the Care and Use of Laboratory Animals (Chonnam National University, Gwangju, Korea). All animal experiments were approved by the Institutional Animal Care and Use Committee (IACUC) of Chonnam National University (Approval No. CNU IACUC-YB-2019-50, CNU IACUC-YB-2017-53), Gwangju, Republic of Korea.

*Prdx5*$^{Ko}$ (B6-*Prdx5*$^{tm1}$/J) mice were gifted by Dr Hyun-ae Woo, Ewha Womans University, Republic of Korea (*Kim et al., 2018*). To obtain the WT and transgenic mice, heterozygous males and females were crossed, and littermates were used for experiments.

Eight-week-old WT and their transgenic female littermates were sham-operated or subjected to bilateral OVX under anesthesia (25 mg/kg of Zoletil and 12.5 mg/kg of Rompun). The mice were sacrificed after 4 weeks, and their serum, uterus, and femurs were collected for biochemical and histo-morphometric analyses.

## Osteoclast and osteoblast differentiation in vitro

Primary mouse pre-osteoblasts were isolated from the calvaria of 3-day-old C57BL/6J mice via sequential digestion with type I collagenase (Gibco) and dispase (Roche), as previously described (*Bellows et al., 1986*). Briefly, the cells were cultured in an α-minimum essential medium (α-MEM), containing 10% characterized heat-inactivated fetal bovine serum (FBS) and 1% penicillin/streptomycin, and differentiated into osteoblasts via treatment with 100 ng/mL of BMP2 (Sino Biological). Cells were harvested at indicated time periods, and ALP staining was performed on day 7. For ALP staining, cells were fixed in 70% ethanol for 1 hr and stained for 10 min with an ALP staining solution (BCIP/NBT alkaline phosphatase kit, Sigma-Aldrich), according to the manufacturer's instructions.

For in vitro osteoclast differentiation, BMM cells were isolated and stimulated with 30 ng/mL of M-CSF (PeproTech) and 50 ng/mL of RANKL (PeproTech), as previously described (*Cho et al., 2021*). To assess the extent of differentiation, the cells were stained using a TRAP kit (Cosmo Bio Co.). The mature osteoclasts were counted under a microscope based on the number of nuclei (n≥3), cell size, and cell number.

**Table 4.** Primer sequences for quantitative reverse transcription-PCR (qRT-PCR).

| Gene | Primer sequence (5' to 3') Forward | Reverse |
|------|---------|---------|
| *Prdx1* | GCATTGAGCAGCCAGAAGAAA | ATCCATCCCCAGCCCTGTAG |
| *Prdx2* | CAATGTGGATGACAGCAAGGA | TTCAGGCTCACCGATGTTTACC |
| *Prdx3* | TGCTGTTGTCAATGGAGAGTTCA | CAAAGGGTAGAAGAAAAGCACCAA |
| *Prdx4* | TTGGTTCAAGCCTTCCAGTACA | ATTATTGTTTCACTACCAGGTTTCCA |
| *Prdx5* | ATTGGATGATTCTTTGGTGTCTCT | CTTCACTATGCCGTTGTCTATCAC |
| *Prdx6* | CCTGATCAGAAAACCGTTGTCA | AGGAAGCATGCCTGTGCAAT |
| *Runx2* | ACTATGGCGTCAAACAGCCT | GGTGCTCGGATCCCAAAAGA |
| *Alpl* | TGGCCTGGATCTCATCAGTATTT | AGTTCAGTGCGGTTCCAGACA |
| *Bglap* | AGAGAGGCAGGGAGGATCAAGT | GGACCTGTGCTGCCCTAAAG |
| *Ctsk* | AGGGAAGCAAGCACTGGATA | GCTGGCTGGAATCACATCTT |
| *Acp5* | CAGCTGTCCTGGCTCAAAA | ACATAGCCCACACCGTTCTC |
| *Fos* | CGAAGGGAACGGAATAAGATG | GCTGCCAAAATAAACTCCAG |
| *Ar* | GACATGCGTTTGGACAGTACCA | TGACAGCCAGAAGCTTCATCTC |
| *Nox1* | CTCCAGCCTATCTCATCCTGAG | AGTGGCAATCACTCCAGTAAGGC |
| *Cybb* | CACAATATTTGTACCAGACAGACTTGAG | AGCTATGAGGTGGTGATGTTAGTGG |
| *Nox4* | CGGGATTTGCTACTGCCTCCAT | GTGACTCCTCAAATGGGCTTCC |
| *Noxo1* | TCAGCAGGTAGCCTGGTTTCCA | CACGGATAGCTCATCAGAGCGA |
| *Cyba* | CCGTCTGCTTGGCCATTG | AACCTGTGGCCGCTCCTT |
| *Ncf4* | AAGACACAGGCAAAACCATCAAG | CTGGAACTCACGCCTCATGA |
| *Ncf1* | TGGTGGTTTTGCCAGATGAA | GCCTCGTCGGGACTGTCA |
| *Ncf2* | tgctcaaggtgcattacaaatacac | CGAGAGCGCCAGCTTCTTAG |
| *Rac1* | GGACACCATTGAGAAGCTGAAGG | GTCTTGAGTCCTCGCTGTGTGA |
| *Rac2* | CCAGCCAAGTGAGGGTCTGA | GAGTGGACAGTCCCAAGAAGGA |
| *18*S | CGCCGCTAGAGGTGAAATTCT | CGAAACTCCGACTTTCGTTCT |

For recovery experiment, pre-osteoblasts from *Prdx5*$^{Ko}$ mice were transfected with pCMV-HA-mPrdx5, pCMV-HA-mPrdx5 C48S, pCMV-HA-mPrdx5 C152S, or pCMV-HA-mPrdx5 C48/152S plasmids with FuGENE-HD (Promega), and then differentiated into osteoblasts with BMP2 stimulation for 7 days. The fixed cells were assayed with an ALP staining solution or ALP assay kit (Takara), according to the manufacturer's instructions.

## Western blot analysis and qRT-PCR

The differentiated osteoblasts and osteoclasts were lysed in a radioimmune assay precipitation buffer (Thermo Fisher Scientific), and western blotting was performed as described previously (*Cho et al., 2021*). Rabbit anti-Prdx1 (Invitrogen), rabbit anti-Prdx2 (Ab Frontier), rabbit anti-Prdx3 (Ab Frontier), rabbit anti-Prdx4 (Abcam), mouse anti-Prdx5 (Invitrogen), rabbit anti-Prdx6 (Invitrogen), mouse anti-AR (Santa Cruz), rabbit anti-hnRNPK (CST), rabbit anti-Lamin β (Ab Frontier), and mouse anti-β-Actin (Sigma-Aldrich) antibodies were used to detect proteins.

Nuclear proteins were isolated from osteoblasts at day 7 under BMP2 stimulation using NE-PER Nuclear and cytoplasmic extraction reagents (Thermo Fisher Scientific) according to the manufacturer's instruction.

Total RNA was extracted using TRIzol reagent (Thermo Fisher Scientific), and cDNA was synthesized as previously described (*Cho et al., 2021*). Quantitative PCR was performed using a SYBR

Green-based system (Thermo Fisher Scientific), and data were calculated using the $2^{-\Delta\Delta CT}$ method. Three separate experiments were performed. The primers used are listed in *Table 4*.

## Micro-CT analysis

Femoral specimens were fixed in a 4% paraformaldehyde solution for 12 hr at 4°C. Micro-CT imaging was performed using a high-resolution Skyscan 1172 system (Bruker-micro-CT, Kontich, Belgium). The images were acquired at a 7 µm voxel resolution, with a 0.5 mm aluminum filter, at 50 kV and 100 µA exposure time, 0.5° rotation, and frame averaging of 1. Image reconstruction software (NRecon; Bruker) was used to reconstruct the serial cross-sectional images using identical thresholds for all samples. To measure the regions of interest (ROIs) of the trabecular and cortical bones, we included ROIs that were 0.7–2.3 mm away from the bottom of the growth plate. The bone morphometric parameters were calculated using adaptive thresholding (the mean of the minimum and maximum values) with CT Analyzer (v1.11.8.0).

## Histology, immunostaining, and dynamic bone histomorphometry

Dynamic bone histomorphometric analysis was performed after injecting 25 mg/kg of calcein (Sigma-Aldrich) or Alizarin Red S (Sigma-Aldrich) into mice as previously described (*Lim et al., 2015*). Briefly, the distal femurs were fixed in a 4% paraformaldehyde solution and subsequently dehydrated with gradient from ethanol to PBS (100%; 95%; 85%; 70; 50%; PBS); the undecalcified femurs were embedded in methyl methacrylate to prepare resin blocks. The resin blocks were cut longitudinally into 6 µm slices of the femur distal metaphysis using a Leica SP1600 microtome (Leica Microsystems, Germany). Fluorescence signals of calcein and Alizarin Red S from the ROIs were captured using a fluorescence microscope (Q500MC, Leica Microsystems). The parameters for dynamic bone histomorphometry were determined using the Bioquant Osteo 2018ME program (Bioquant Osteo, Nashville, TN, USA).

Goldner's trichrome staining was performed on 3 µm long paraffin-embedded sections. After rehydration, the slides were washed in distilled water, refixed in Bouin's solution (Sigma-Aldrich) for 15 min at 56°C, and rinsed with running tap water for 5 min to remove picric acid (yellow). The slides were then counterstained with Weigert's hematoxylin (Sigma-Aldrich) for 10 min, washed with tap water for 5 min, and rinsed thrice with distilled water. Next, they were stained with Biebrich scarlet-acid fuchsin (Sigma-Aldrich) for 5 min and rinsed thrice with distilled water. Afterward, they were immersed in phosphotungstic/phosphomolybdic acid (Junsei) for 10 min and transferred to aniline blue solution (Junsei) for 5 min. Finally, the slides were washed with distilled water and treated with 1% acetic acid for 1 min. After dehydration and mounting, the stained bone sections were observed under a microscope (Q500MC, Leica Microsystems), and the parameters of osteoid volume/bone volume were determined using the Bioquant Osteo 2018ME program (Bioquant Osteo).

Osteoclasts and osteoblasts were visualized using TRAP and ALP staining, respectively. TRAP (TRAP Staining Kit, Cosmo Bio Co.) staining was carried out according to the manufacturer's instructions, with some modifications. NBT/BCIP staining (Sigma-Aldrich) was carried out by incubating tissue sections. The sections were then counterstained with 0.05% Fast Green FCF (Sigma-Aldrich), dehydrated using graded ethanol solutions, and allowed to dry without clearing in xylene before mounting. Positive cells were visualized by purple color and analyzed using the ImageJ software (https://imagej.nih.gov/ij/ v1.53o).

## Enzyme-linked immunosorbent assay

The levels of specific markers of osteogenesis in the serum from WT and *Prdx5*$^{Ko}$ mice at 12 weeks were measured using enzyme-linked immunosorbent assay (ELISA) according to the manufacturer's description. The RANKL levels were measured using a mouse RANKL ELISA kit (Abcam); the OPG levels were measured using a Quantikine ELISA (R&D Systems) kit; and the BMP2 levels were measured using mouse BMP2 ELISA kits, respectively (LSBio).

The testosterone levels in the serum from WT and *Prdx5*$^{Ko}$ male mice at 4, 8, and 12 weeks were measured using the testosterone ELISA kit (R&D Systems) according to the manufacturer's description.

## Measurement of intracellular ROS levels

For osteoblasts, calvarial cells from WT and *Prdx5*[Ko] mice were cultured for two days in a medium containing BMP2. For osteoclasts, BMMs from WT and *Prdx5*[Ko] mice were cultured for 2 days in a medium containing M-CSF and RANKL. The cells were washed with α-MEM lacking phenol red and then incubated with 10 μM of CM-H$_2$DCFDA (Thermo Fisher Scientific) for 30 min. Fluorescence intensity was measured using a multiplate reader (SpectraMax i3x, Molecular Devices) and visualized under a microscope (Olympus Corp., IX2-ILL100) at excitation and emission wavelengths of 490 and 520 nm, respectively.

## Calvarial bone defect models and micro-CT analysis

For the calvarial bone defect model, a critical size calvarial defect was created using a 5 mm diameter trephine bur (Fine Science Tools, Foster City, CA, USA) and covered with absorbable collagen sponges containing 300 ng of BMP2 (Cowell Medi Corp., Seoul, Republic of Korea) in 12-week-old *Prdx5*[Ko] and WT C57BL6/J male mice. After 3 weeks, the model mice were sacrificed for analysis. Briefly, the mice were subjected to inhalational anesthesia using an XGI-8 Gas Anesthesia System (PerkinElmer, Waltham, MA, USA) containing a mixture of 4% isoflurane (ISOTROY 100, Troikaa, India) and oxygen, for 4 min. The osteological structures of the specimens were examined using a micro-CT scanning system, combined with a Quantum GX μCT imaging system (PerkinElmer), at the Korea Basic Science Institute (Gwangju, Republic of Korea). The scanned skeletal data were reconstructed into 3D tomograms comprising high-contrast images of the skeletal parts of interest.

## Confocal microscopy

The cells were grown on sterilized glass coverslips and fixed in 4% paraformaldehyde. Non-specific binding was blocked by incubating slides in 0.1% bovine serum albumin in PBS. Subsequently, the samples were stained with mouse anti-Prdx5 (1:200, Invitrogen) and rabbit anti-hnRNPK antibodies (1:200, Cell Signaling Technology), followed by incubation with Alexa 555- or Alexa 488-conjugated secondary antibodies (1:500, Invitrogen) and DAPI/antifade (1:200, Invitrogen). Images were captured using a confocal laser scanning microscope equipped with visible and near-infrared lasers. Images were acquired using the Airyscan mode supported by the LSM 880 confocal laser scanning microscope for image optimization (Carl Zeiss, Oberkochen, Germany).

## Immunoprecipitation

Pre-osteoblasts isolated from mouse calvaria were cultured for 7 days in a BMP2-containing or normal medium (CTRL). The cells were lysed with an IP lysis buffer (150 mM NaCl, 25 mM Tris-HCl, 10% glycerol, and 1 mM EDTA) containing a protease inhibitor cocktail (Roche, Basel, Switzerland). The lysed cells were centrifuged, and equal amounts of proteins were incubated with an anti-Prdx5 antibody, or an IgG rabbit polyclonal antibody (Cell Signaling Technology) as a negative control. The proteins were further incubated with protein A/G-sepharose beads (GE Healthcare) for 2 hr. The beads were then washed five times with a lysis buffer to remove the immunocaptured proteins, boiled, and then subjected to western blot analysis using anti-Prdx5 (1:500, Ab Frontier) and anti-hnRNPK (1:500, Cell Signaling Technology) antibodies.

## LC–MS/MS analysis

Sodium dodecyl sulfate–polyacrylamide gel electrophoresis and in-gel digestion were performed as previously described (*Yun et al., 2018*). Briefly, the sliced gel was digested in trypsin gold (Promega, Madison, WI, USA). The tryptic peptides solution was dried using a speed vacuum concentrator for LC–MS/MS (*Lee et al., 2016*). Briefly, tryptic digested samples were dissolved with 0.5% trifluoroacetic acid before further analysis. A 5 μL dissolved sample was loaded onto an MGU-30 C18 trapping column (LC Packings, Amsterdam, The Netherlands). The concentrated tryptic peptides were eluted from the column and loaded directly into a 10 cm ×75 μm ID C18 reverse phase column at a flow rate of 300 nL/min. The peptides underwent gradient elution in 0–55% acetonitrile over 100 min. MS and MS/MS spectra were acquired in the data-dependent mode using the LTQ-Velos ESI ion trap mass spectrometer (Thermo Fisher Scientific). For protein identification, the MS/MS spectra were analyzed with MASCOT v2.4 (Matrix Science, UK) using the mouse protein database downloaded from Uniprot. The mass tolerance for the parent or fragmentation was 0.8 Da. Carbamidomethylation

of cysteine and oxidation of methionine were considered in MS/MS analysis as variable modifications of the tryptic peptides. The MS/MS data were filtered according to a false discovery rate criterion of 1%. Each sample was analyzed in triplicate. For protein quantification, we used the mol% value, which was calculated from the emPAI values in the MASCOT program (*Lee et al., 2016*; *Yun et al., 2018*). The canonical pathway of Prdx5-interacting proteins was screened using Ingenuity Pathway Analysis (IPA, Ingenuity Systems, Redwood City, CA, USA, https://www.ingenuity.com), which leverages the Ingenuity Knowledge Base. Protein–protein interactions were constructed using STRING v11 (*Szklarczyk et al., 2019*).

## Luciferase reporter assays

MC3T3-E1 cells were cultured in α-MEM containing 10% FBS and 1% penicillin–streptomycin. Cell line identity was confirmed by PCR amplification methods from Korean Cell Line Bank (KCLB). Cells were transiently transfected with pGL3-*Bglap* promoter-Luc reporter using Lipofectamine 3000 (Invitrogen). The transfection efficiency was determined by co-transfecting the cells with a beta-galactosidase reporter (SV-β-gal). The reporter vectors were obtained from professor Won-Gu Jang, Daegu University, South Korea. The cells were transfected again with scrambled siRNA, Prdx5 siRNA, or pCMV-HA-Prdx5 plasmids. After the cells were recovered, osteoblast differentiation was induced by incubating them with 200 ng/mL of BMP2 for 72 hr. Luciferase activity was measured using a luciferase reporter assay system (Promega) and a luminometer (SpectraMax i3x, Molecular Devices) according to the manufacturer's instructions. The experiments were performed in triplicate and repeated thrice.

## ChIP assays

For the ChIP assay, the Cell Signaling ChIP assay was used according to the manufacturer's description. We searched the 1500 bp upstream promoter region of *Bglap* in the National Center for Biotechnology Information (NCBI) and the published manuscript (*Stains et al., 2005*) to predict the DNA-binding sites for hnRNPK. Briefly, the osteoblasts within BMP2 for 7 days and pre-osteoblasts from WT and *Prdx5*$^{Ko}$ mice were crosslinked, quenched, and sonicated to extract chromatin. For IP, rabbit anti-hnRNPK antibody (CST) or normal rabbit IgG (CST) was used. PCR was performed with following primers: F: 5'- TTTGACCCACTGAGCACATGA-3' and R: 5'-GACTTGTCTGTTCTGCACCC -3'. Whole chromatin without IPs was used as input DNA templates.

## RNA-seq analysis

BMMs were cultured for 4 days in an M-CSF and RANKL-containing medium for differentiating them into osteoclasts, and then lysed for RNA extraction. RNA was isolated using the RNeasy Mini Kit (Qiagen, Hilden, Germany), and quality control and sequencing were performed by Macrogen Inc (Seoul, Republic of Korea). Briefly, a cDNA library was prepared using the TruSeq Stranded mRNA LT Sample prep kit (Illumina Inc), and cDNA was synthesized using SuperScript II reverse transcriptase (Thermo Fisher Scientific).

All raw sequence reads were preprocessed using Trimmomatic (v0.39) (*Bolger et al., 2014*) to remove adapter sequences and bases with low sequencing quality. The remaining clean reads were mapped based on the mouse reference genome (mm10) using Hisat2 (v2.1.0) (*Kim et al., 2015*) with the default parameters. BAM files generated by HiSat2 were further processed with Cufflinks (v2.2.1) (*Trapnell et al., 2012*) to quantify transcript abundance using the fragment per kilobase of exon per million fragments mapped (FPKM) normalization. Differential expression was analyzed using Cuffdiff (v2.2.1) to identify DEGs with FPKM >1 in at least one sample and q-value <0.05. We performed enrichment analysis of GO categories using the DAVID functional annotation tool (https://www.david.ncifcrf.gov). The mouse reference genome sequence and annotation data were downloaded from the UCSC genome browser (https://www.genome.ucsc.edu), and the R software was used to visualize the results.

## Statistics

Each experiment with cells was repeated at least thrice. Data are presented as mean ± standard error of the mean (SEM) or standard deviation (SD). The statistical analysis tests performed were a two-tailed Student's t-test. Image-based data were analyzed using the GraphPad Prism statistical software. Differences were considered statistically significant at *p < 0.05 and **p < 0.01.

## Acknowledgements

We greatefully acknowledge Korea Mouse Phenotyping Center (KMPC) for technical support of mouse phenotyping.

## Additional information

### Funding

| Funder | Grant reference number | Author |
|---|---|---|
| Ministry of Science and ICT (South Korea) | 2014M3A9D5A01073658 | Je-Yong Choi |

The funders had no role in study design, data collection and interpretation, or the decision to submit the work for publication.

### Author contributions

Eunjin Cho, Conceptualization, Resources, Data curation, Formal analysis, Validation, Writing - original draft; Xiangguo Che, Resources, Data curation, Formal analysis; Mary Jasmin Ang, Jinkyung Lee, Resources, Data curation; Seongmin Cheon, Software, Visualization; Kwang Soo Kim, Chang Hoon Lee, Conceptualization, Data curation; Sang-Yeop Lee, Resources, Software; Hee-Young Yang, Data curation; Changjong Moon, Methodology; Chungoo Park, Software, Methodology; Je-Yong Choi, Conceptualization, Funding acquisition; Tae-Hoon Lee, Conceptualization, Funding acquisition, Project administration, Writing - review and editing

### Author ORCIDs

Eunjin Cho http://orcid.org/0000-0003-0276-4728
Chang Hoon Lee http://orcid.org/0000-0001-8953-9069
Changjong Moon http://orcid.org/0000-0003-2451-0374
Tae-Hoon Lee http://orcid.org/0000-0003-0105-1750

### Ethics

All animals were housed in a specific pathogen-free facility following the guidelines provided in the Guide for the Care and Use of Laboratory Animals (Chonnam National University, Gwangju, Korea). All animal experiments were approved by the Institutional Animal Care and Use Committee (IACUC) of Chonnam National University (Approval No. CNU IACUC-YB-2019-50, CNU IACUC-YB-2017-53), Gwangju, Republic of Korea.

### Decision letter and Author response

Decision letter https://doi.org/10.7554/eLife.80122.sa1
Author response https://doi.org/10.7554/eLife.80122.sa2

## Additional files

### Supplementary files
• MDAR checklist

### Data availability

Proteomics data that support the findings of the current study have been deposited to the ProteomeXchange Consortium via the PRIDE (*Perez-Riverol et al., 2019*) partner repository with the dataset identifiers PXD020082 and http://doi.org/10.6019/PXD020082. RNA-seq data that support the findings of this study are available at the Gene Expression Omnibus (GEO) site of the NCBI (https://www.ncbi.nlm.nih.gov/sra/PRJNA848442).

The following datasets were generated:

| Author(s) | Year | Dataset title | Dataset URL | Database and Identifier |
|---|---|---|---|---|
| Lee S-Y | 2020 | Mouse (C57BL6/J) osteoblast LC-MS/MS | http://doi.org/10.6019/PXD020082 | PXD020082, 10.6019/PXD020082 |
| Cho E | 2022 | RNA-seq analysis on mouse bone marrow derived macrophages and osteoclasts from control and Prdx5 knockout mice | https://www.ncbi.nlm.nih.gov/bioproject/PRJNA848442/ | NCBI BioProject, PRJNA848442 |

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
