## [Editor Report]

Peroxiredoxin 5 regulates osteogenic differentiation via interaction with hnRNPK during bone regeneration is an important study as the fundamental role of Peroxiredoxin 5 has been established in bone regeneration. The study is compelling with experimentally establishing the role of Peroxiredoxin- 5 in osteoblast, and osteoclast along with in vivo studies using Prdx5 knockout mice to establish the functional role of Prdx in bone homeostasis and further as a therapeutic target.

---

## [Decision Letter]

**Decision letter after peer review:**

Thank you for submitting your article "Peroxiredoxin 5 regulates osteogenic differentiation via interaction with hnRNPK during bone regeneration" for consideration by *eLife*. Your article has been reviewed by 2 peer reviewers, and the evaluation has been overseen by a Reviewing Editor and Mone Zaidi as the Senior Editor. The following individuals involved in review of your submission have agreed to reveal their identity: Aicha Asma Houfani (Reviewer #2).

Essential revisions

The reviewers have discussed their reviews with one another, and the Reviewing Editor has drafted this to help you prepare a revised submission. Please address all recommendations from reviewers and supply a point-by-point rebuttak.

*Reviewer #1 (Recommendations for the authors):*

The manuscript contains several novel and interesting findings, including positive

regulation of osteoblast differentiation by Prdx5, while the effect is opposite in osteoclasts; the induction of Prdx5 expression by BMP2 in osteoblasts; the demonstration of the osteoporotic phenotypes in Prdx5KO male mice; the interaction between Prdx5 and hnRNPK, associated with the osteoblast differentiation and bone remodelling. However, these novel observations are not well integrated into a cohesive and well-argued story, as knowledge gaps and inappropriate interpretation of generated results are notable.

1. The authors should avoid ambiguity in the manuscript by stating that osteoporotic phenotypes are observed in Prdx5KO mice, as the observed changes are only only observed in male mice.

2. Prdx5 is known to be localized to mitochondria, peroxisomes and the cytosol. The authors describe BMP2-induced translocation of Prdx5 from the cytosol to the nucleus. They also report the interaction between transiently overexpressed Prdx5 and hnRNPK in osteoblasts. The obvious and unexplored question remains whether BMP2 treatment can induce the interaction between transiently overexpressed and endogenous Prdx5 and hnRNPK.

3. The role of Prdx5 in maintaining ROS homeostasis is well established. It remains to be investigated whether the Prdx5 peroxidase activity is associated with osteoblast differentiation. Transient or stable overexpression of wild type or peroxidase deficient mutant of Prdx5 in Prdx5KO pre-osteoblasts in the presence or absence of BMP2 may address this question.

4. Recently, J. Du et al. showed that the expression of Prdx5 in osteoblasts is upregulated after the ovariectomy. How these findings reconcile with the fact that the ovariectomy revealed no significant differences between WT and Prdx5KO mice on bone remodelling.

J. Du et al. Ovariectomy upregulated the expression of Peroxiredoxin 1&5 in osteoblasts of mice. Sci Rep. 2016 Oct 27;6:35995.

5. Figures and legends should be more clearly presented. For example, lower panels of qRT-PCR analysis in Figure 2A and 2D are not properly labelled – what are the differences between lower panels?

6. In Discussion, the role of NADPH oxidases (NOXs) in BMP2-induced ROS production should be discussed and how this relates to the role of Prdx5 in

responding to increased ROS production.

There are also some inaccuracies in the manuscript:

Page 3. The authors incorrectly state that "Prdxs are classified as 1-Cys (Prdx1-5) and 2-Cys (Prdx6) based on their conserved cysteine residues". It is vice versa.

Page 28. Six members of the Prdx family reportedly exhibit antioxidant activities owing to the presence of CXXC amino acid sequences (Chae et al., 1994; Rhee et al., 2001). Incorrect statement – Prdx6 possesses only one Cys in the catalytic pocket.

In summary, major revisions are required for the manuscript to be suitable for publication in *eLife*.

*Reviewer #2 (Recommendations for the authors):*

Line 68: There could be a transition sentence here to explain how heterogeneous nuclear ribonucleoproteins are related to peroxiredoxins in the previous paragraph.

Line 84: To make reading easier, define and indicate what c-myc and c-src are.

Line 122-123: provide a correlation statistical value here.

Figure 1: Please include the composition of the growth and osteo media in the corresponding method section.

Figure 2, A and D: make it clear what the right and left bar plots represent.

Figure 4, D: indicate what represents the y axis.

Line 372: correct raged to ranging

Line 412: Figure 7—figure supplement 1. Correct to " a total of 153 "say what is" downregulated.

Line 443: Be explicit about the degree of the bone mass decrease and its significance.

Line 455-456: What the writers mean by "scavenged" in this context is unclear to me.

Line 570: Add percentage of ethanol used.

Line 572: Add version of the software.

Line 616: Why specifically 7 days?

Line 621: Do you have an image of the gel? Have you sliced single bands or the whole gel lanes?

Line 622: Specify whether you used Trypsin-ultra Mass Spectrometry Grade and the trypsin's name manufacturer.

"The tryptic peptides were dried and extracted for LC-MS/MS"

The samples have been peptide extracted before they have been peptide digested, thus I would change the word extracted here.

Before getting to the mass spec, were the samples C18 cleaned? What equipment and MS methods have you applied?

I think this part should move to line 634 where you talked about C18 before drying samples and sending them to MS.

Paragraph 623-629: I guess this part should come before you inject the samples to the MS?

Line 631: Rearrange your method description so that the analyses come immediately after the section in which you first discussed LC-MS/MS, at 622.

The protein of interest belongs to the peroxiredoxins (Prxs) family of cysteine-dependent peroxidase enzymes. Have you reduced and alkylated your samples? I noticed that reduction and alkylation weren't included. I wonder if this was done on purpose to prevent interference with the protein of interest, or if the authors simply omitted to add the steps? Also, have you used an antioxidant when running your gel?

---

## [Author Response]

Reviewer #1 (Recommendations for the authors):The manuscript contains several novel and interesting findings, including positiveregulation of osteoblast differentiation by Prdx5, while the effect is opposite in osteoclasts; the induction of Prdx5 expression by BMP2 in osteoblasts; the demonstration of the osteoporotic phenotypes in Prdx5KO male mice; the interaction between Prdx5 and hnRNPK, associated with the osteoblast differentiation and bone remodelling. However, these novel observations are not well integrated into a cohesive and well-argued story, as knowledge gaps and inappropriate interpretation of generated results are notable.1. The authors should avoid ambiguity in the manuscript by stating that osteoporotic phenotypes are observed in Prdx5KO mice, as the observed changes are only only observed in male mice.

We apologize for this oversight. We have modified the manuscript to clearly indicate that osteoporotic phenotypes were observed in *Prdx5 KO* male mice.

2. Prdx5 is known to be localized to mitochondria, peroxisomes and the cytosol. The authors describe BMP2-induced translocation of Prdx5 from the cytosol to the nucleus. They also report the interaction between transiently overexpressed Prdx5 and hnRNPK in osteoblasts. The obvious and unexplored question remains whether BMP2 treatment can induce the interaction between transiently overexpressed and endogenous Prdx5 and hnRNPK.

Thank you for mentioning the point that we missed to include in the manuscript. Yes, the interaction between hnRNPK and Prdx5 is induced by BMP2 treatment, which is a stimulator of osteoblast differentiation. When we examined the localization of Prdx5, we noted that BMP2 stimulation was responsible for its translocation to the nucleus (Figure 5—figure supplement 1). We assume that hnRNPK accumulates on the *Ocn* promoter and inhibits its expression in the nucleus without BMP2 stimulation. When BMP2 induces Prdx5 translocation into the nucleus, Prdx5 removes hnRNPK from the *Ocn* promoter by interacting with the latter, followed by *Ocn* transcription. To support our hypothesis, we performed ChIP assay on the OG2 (*Ocn*) promoter. We have included the revised data in Figure 6—figure supplement 1 and the Results sections. Indeed, we have included the summary model in Figure 8.

3. The role of Prdx5 in maintaining ROS homeostasis is well established. It remains to be investigated whether the Prdx5 peroxidase activity is associated with osteoblast differentiation. Transient or stable overexpression of wild type or peroxidase deficient mutant of Prdx5 in Prdx5KO pre-osteoblasts in the presence or absence of BMP2 may address this question.

Thank you for your useful suggestion. We also wanted to investigate the antioxidant function of Prdx5; therefore, we determined the ROS levels (Figure 2—figure supplement 2). However, the ROS levels were even lower in *Prdx5 KO* than those in WT, while they increased by BMP2 stimulation in WT. We assumed that other peroxiredoxins can compensate for the deficient effect of Prdx5. Also, because all Prdxs are oxidized by hydrogen peroxide and reduced by thioredoxin, their compensative effect is reasonable.

We developed a cysteine mutant at Cys48 and/or Cys152 of Prdx5 to transfect *Prdx5 KO* osteoblast cells transiently, following the reviewer’s suggestion. We compared osteoblast differentiation by ALP assays (Figure 2—figure supplement 3); no major differences were observed between the mutants and WT-transfected cells. Because of the poor transfection efficiency in the primary cells, we tried to develop stable overexpression cells; however, culturing for long periods in primary cells remains a challenge. Therefore, we concluded that Prdx5 peroxidase activity is not significantly associated with osteoblast differentiation in our study.

4. Recently, J. Du et al. showed that the expression of Prdx5 in osteoblasts is upregulated after the ovariectomy. How these findings reconcile with the fact that the ovariectomy revealed no significant differences between WT and Prdx5KO mice on bone remodelling.J. Du et al. Ovariectomy upregulated the expression of Peroxiredoxin 1&5 in osteoblasts of mice. Sci Rep. 2016 Oct 27;6:35995.

Thank you for your thoughtful comments. The first difference between the studies of J. Du et al. and ours was mouse strain. J. Du et al. used Kunming mice, while we used C57BL6/J mice. In addition, Du et al. detected the Prdx5 expression in osteoblasts (Figure 4 in Du et al.). We also reported a similar result, that is Prdx5 expression was high in osteoblasts. However, Du et al. did not clearly indicate osteoblast and osteoclast cells in the trabecular bone. We suggested that the Prdx5 expression increased in the osteoblasts of OVX as a protective effect against bone loss. However, the amount of other antioxidants such as Prdx1 also increased in their results. Therefore, osteoporosis phenotypes may not be observed only in Prdx5 KO mice induced by OVX, which would be observed in double KO. If we get a suitable opportunity, we plan to further investigate Prdx1 and Prdx5 double KO mice in a future study on osteoporosis.

6. In Discussion, the role of NADPH oxidases (NOXs) in BMP2-induced ROS production should be discussed and how this relates to the role of Prdx5 in responding to increased ROS production.

Thank you for such a useful suggestion. We have tested the Nox expression levels in both osteoblasts and osteoclasts (Figure 2—figure supplementary 2). We have also included the information on Nox-mediated ROS generation in osteoblasts in the Discussion section.

Reviewer #2 (Recommendations for the authors):Introduction: There could be a transition sentence here to explain how heterogeneous nuclear ribonucleoproteins are related to peroxiredoxins in the previous paragraph.

Thank you for your useful suggestion. However, the correlation between hnRNPs and Prdxs has been poorly studied. Although hnRNPK binds to antioxidant *NFE2L2* transcripts encoding Nrf2 antioxidant transcription factor in ALS patients (DOI: 10.1093/hmg/ddx093), our study is the first to suggest a correlation between Prdx5 and hnRNPK. To make it clearer, we have deleted/moved the information on hnRNP from the Introduction section and modified the section on hnRNPs in Discussion.

Western blot analysis and qRT-PCR: Why specifically 7 days?

Mesenchymal stromal cells from the neonatal mouse calvariae differentiate into osteoblasts by BMP2 stimulation in 4–21 days (DOI: 10.1016/j.bone.2009.09.019). On day 7, osteoblast-related genes, including *Alp*, *Runx2*, and *Ocn*, were expressed, and ALP-positive cells were detected as shown in Figure 2. Therefore, we collected the cells on day 7 after BMP2 stimulation to determine the Prdx5-interacting proteins in osteoblast cells, not in pre-osteoblast cells.

Immunoprecipitation: Do you have an image of the gel? Have you sliced single bands or the whole gel lanes?

We apologize that we do not have the image. We directly sliced the gel based on the bands (molecular weights/size) and digested directly.

Immunoprecipitation: Specify whether you used Trypsin-ultra Mass Spectrometry Grade and the trypsin's name manufacturer."The tryptic peptides were dried and extracted for LC-MS/MS"The samples have been peptide extracted before they have been peptide digested, thus I would change the word extracted here.Before getting to the mass spec, were the samples C18 cleaned? What equipment and MS methods have you applied?I think this part should move to line 634 where you talked about C18 before drying samples and sending them to MS.Immunoprecipitation: I guess this part should come before you inject the samples to the MS?LC-MS/MS analysis: Rearrange your method description so that the analyses come immediately after the section in which you first discussed LC-MS/MS, at 622.

Thank you for the detailed comments. Based on your suggestions, we have modified the Methods section by rearranging the text and adding more detailed information. For the LC-MS/MS instrument, we used two columns (MGU-30 C18 and reverse phase columns) and directly loaded the samples onto the columns, thus not going through a C18 cleaning process. The LC-MS/MS data were generated using a data-dependent mode with the help of an LTQ-Velos instrument.

The protein of interest belongs to the peroxiredoxins (Prxs) family of cysteine-dependent peroxidase enzymes. Have you reduced and alkylated your samples? I noticed that reduction and alkylation weren't included. I wonder if this was done on purpose to prevent interference with the protein of interest, or if the authors simply omitted to add the steps? Also, have you used an antioxidant when running your gel?

Our purpose was simply to determine the interacting proteins of Prdx5. During osteoblast differentiation, we noticed that Prdx5 is localized in the nucleus. Therefore, we determined the proteins that guide Prdx5 into the nucleus or binding partners in the nucleus, which had not been studied so far. We did not focus on the antioxidant function of Prdx5 in the nucleus. Therefore, we did not include reduction steps or addition of antioxidants in the gel. In addition, we did not want to modify the status of Prdx5 by adding antioxidants. Note that we performed tandem mass spectrometry analysis by oxygen quenching to determine the antioxidant function of Prdx5 in our previous study (DOI: 10.1021/pr100190b).